# Soil moisture dominates dryness stress on ecosystem production globally

Laibao Liu [1,2✉], Lukas Gudmundsson [1], Mathias Hauser [1], Dahe Qin[2], Shuangcheng Li[2] & Sonia I. Seneviratne [1✉]

Dryness stress can limit vegetation growth and is often characterized by low soil moisture (SM) and high atmospheric water demand (vapor pressure deficit, VPD). However, the relative role of SM and VPD in limiting ecosystem production remains debated and is difficult to disentangle, as SM and VPD are coupled through land-atmosphere interactions, hindering the ability to predict ecosystem responses to dryness. Here, we combine satellite observations of solar-induced fluorescence with estimates of SM and VPD and show that SM is the dominant driver of dryness stress on ecosystem production across more than 70% of vegetated land areas with valid data. Moreover, after accounting for SM-VPD coupling, VPD effects on ecosystem production are much smaller across large areas. We also find that SM stress is strongest in semi-arid ecosystems. Our results clarify a longstanding question and open new avenues for improving models to allow a better management of drought risk.

[1] Institute for Atmospheric and Climate Science, ETH Zurich, Zurich, Switzerland. [2] College of Urban and Environmental Sciences, Peking University, Beijing, China. ✉email: laibao.liu@env.ethz.ch; sonia.seneviratne@ethz.ch

L ow soil moisture (SM) supply and high atmospheric water demand (vapor pressure deficit, VPD) are considered as the two main drivers of dryness stress on vegetation, which can cause large threats to agricultural production[1] and drive widespread tree mortality[2]. Recently, it has also been shown that the capacity of land ecosystems to act as a future carbon sink is highly dependent on the influence of SM on ecosystem carbon fluxes[3]. Accurate understanding of dryness stress on ecosystems is therefore critical to manage drought risks and to reduce uncertainties in predicting future land carbon uptake and climate change.

However, there is an ongoing debate on the relative role of SM and VPD in determining the response of vegetation to dryness, leading to divergent assessments of dryness stress on plant carbon uptake in the scientific literature, as well as in their representation in models. On the one hand, SM is the direct water pool of plants and determines the amount of water that can be extracted by plant roots. Thus, low precipitation or SM availability are most commonly used to identify vegetation dryness stress and are well documented to successfully capture the consequences of dryness on vegetation productivity[4–6], also resulting in feedbacks of plants' activity to climate[7–9]. On the other hand, high VPD may induce plants to close stomata to minimize water loss at the leaf scale[10], and is expected to constrain plant photosynthesis in ecosystems. Some recent studies emphasize the importance of VPD and suggest that it may have stronger effects than SM in determining ecosystem water and carbon fluxes[11,12]. However, the relative role of low SM and high VPD in limiting vegetation productivity at the ecosystem scale remain unclear. As a consequence, in combination with the uncertainty in physiological process understanding, dryness stress on photosynthesis is either represented as a function of SM only[13,14], VPD only[15–17], or both[18] in terrestrial ecosystem models (TEMs) and satellite models. For instance, the TEM JSBACH does not incorporate a stomatal response to VPD[19], because it is uncertain if the SM-VPD correlation will cause a double counting of the dryness sensitivity. In contrast, in the TEM G'Day, VPD can limit plant photosynthesis by causing stomatal closure, and SM can constrain plant photosynthesis directly[19].

Here, with simultaneous use of several independent satellite observations of solar-induced chlorophyll fluorescence (SIF) and climate data sets, we first decouple the strong correlations between SM and VPD and then disentangle their respective effects in limiting ecosystem production globally. Our results demonstrate that SM has a dominant role in determining ecosystem production dryness stress over most land vegetated areas compared with that of VPD.

## Results and discussion
**Coupling of SM and VPD confounds ecosystem dryness stress**. The difficulty to disentangle the respective effects of SM and VPD stems from the fact that SM and VPD are strongly coupled through land–atmosphere interactions[7,20]. In addition, field experiments that manipulate atmospheric humidity and temperature at the ecosystem scale are lacking[21]. Given the strong SM-VPD coupling (Fig. 1c), e.g., on the yearly scale, both lower SM and higher VPD are associated with lower ecosystem gross primary production (GPP), indicated by SIF (Fig. 1a, b). This underlies the use of either SM or VPD alone as proxy for dryness stress on ecosystem production in many current models. Note a global spatially contiguous SIF data set was mainly used in this study, which was generated by using the machine-learning algorithm to train SIF observations from Orbiting Carbon Observatory-2 (OCO-2)[22]. We display the yearly scale because it is typically used to represent the condition of strong SM-VPD

coupling globally[11], and the study time period mainly spans from 2001 to 2016. However, as SM and VPD are strongly coupled, it is possible that the correlation between SM and SIF is a byproduct of the correlation between VPD and SIF, or vice versa. As a consequence of SM-VPD coupling, the correlations of yearly SM and VPD with SIF is very similar globally (Fig. 1d). Consequently, the correlation between SM and VPD constitutes a confounding factor that is often overlooked when assessing the role of SM and VPD in determining the impact of dryness stress on ecosystem production. There are still low correlations between SIF and SM or VPD in the northern high latitudes or tropical regions, which suggests possible temperature or radiation effects and requires further investigation.

**Decoupling of SM and VPD globally**. At yearly scale, there is a strong negative correlation between SM and VPD, indicating that low SM is always accompanied by high VPD (Fig. 1c), which is consistent with previous findings[7,20]. From yearly to monthly, weekly, and daily scale, the correlations between SM and VPD are generally decreasing (Fig. 2d), but remain large across extensive areas, such as central South America, Sub-Saharan Africa, India, and Southeast Asia (Fig. 2a and Supplementary Fig. 1). However, when binning the data into 10 bins according to percentiles of either SM or VPD per pixel, we find that the correlation coefficient between SM and VPD in each bin becomes approximately zero (Fig. 2b–d and Supplementary Figs. 2 and 3). This shows that SM and VPD are generally decoupled at daily scale in both SM and VPD bins.

**Disentangling the relative role of SM of VPD**. We now disentangle the respective effects of SM and VPD in limiting ecosystem production by exploiting the fact that SM and VPD are decoupled in binned daily SM or VPD data (Fig. 2). SM and VPD are also largely decoupled in 4-day bins, which is the temporal resolution of the mainly used SIF data set (Supplementary Figs. 4 and 5). The analysis is guided by the assumption that if SM dominates dryness stress, low SM will limit ecosystem production regardless of VPD variations (Supplementary Fig. 6a, c). In the same way, if VPD dominates dryness stress, high VPD will limit ecosystem production regardless of SM variations (Supplementary Fig. 6b, d).

To illustrate this further, we select an example pixel located in Mali (West Africa). Without decoupling SM and VPD, it is difficult to conclude whether the decrease in SIF is caused by low SM, high VPD, or both in conjunction (Fig. 3a, b). However, when looking at the variation of SIF across VPD gradients in SM bins (without SM-VPD coupling), high VPD does not reduce SIF but even increase SIF a bit under moderate SM conditions (Fig. 3c). In contrast, low SM reduces SIF noticeably in VPD bins (Fig. 3d). This shows that high VPD does not limit SIF in the absence of the SM-VPD coupling at the example pixel, whereas low SM can still limit SIF. In other words, the apparent VPD limitation on SIF is largely the byproduct of SM-VPD coupling. The respective effects of SM and VPD on SIF is also illustrated in Fig. 3e. The changes in SIF from low VPD to high VPD without SM-VPD coupling (termed ΔSIF(VPD|SM)) can quantify the VPD stress on SIF. Likewise, changes in SIF from high SM to low SM without SM-VPD coupling (termed ΔSIF(SM|VPD)) quantify the SM stress on SIF. The effect of SM and VPD on SIF is estimated using two approaches: (i) SIF in the maximum VPD bin minus SIF in the minimum VPD bin or SIF in the minimum SM bin minus SIF in the maximum SM bin; (ii) using linear regression to derive changes in SIF caused by high VPD or low SM. The two approaches lead to similar results (Methods and Supplementary Fig. 16). As shown in Fig. 3f, the SM effect is strong at the example location (ΔSIF(SM|VPD) = −0.17 mW

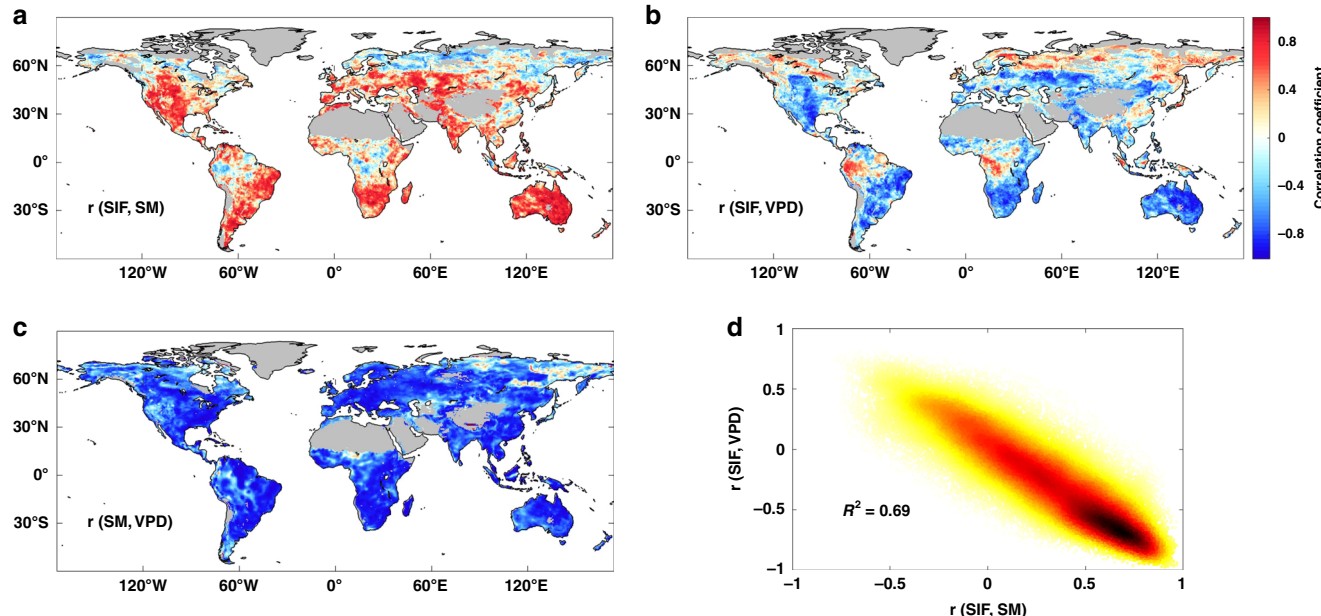

**Fig. 1 Strong coupling of soil moisture and vapor pressure deficit confounds ecosystem dryness stress. a–c** Spatial distribution of Pearson's correlation coefficient between solar-induced chlorophyll fluorescence (SIF) and soil moisture (SM) (r(SIF, SM)), SIF and vapor pressure deficit (VPD) (r(SIF, VPD)), and SM and VPD (r(SM, VPD)), at the yearly scale. Regions with sparse vegetation and regions without valid data are masked in gray. **d** Relationship between yearly r(SIF, VPD) and yearly r(SIF,SM) across land vegetated areas. Color shows the relative density of data points, with higher density in black and lower density in yellow.

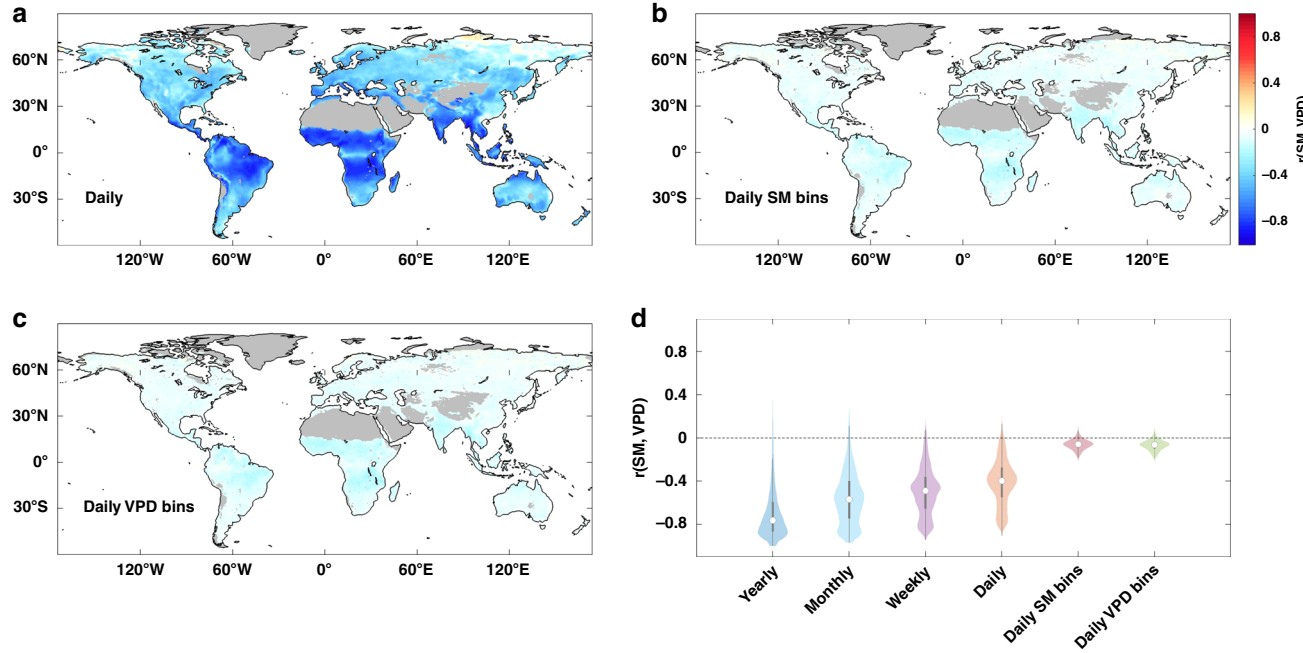

**Fig. 2 Decoupling of soil moisture and vapor pressure deficit. a–c** Spatial distribution of Pearson's correlation coefficient between soil moisture (SM) and vapor pressure deficit (VPD) at daily scale, averaged over daily SM bins, and averaged over daily VPD. Regions with sparse vegetation and regions without valid data are masked in gray. **d** Violin plots of correlations between SM and VPD from yearly to daily bins across land vegetated areas. White dots indicate the median values, gray boxes cover the interquartile range, and thin gray lines reach the 5th and 95th percentiles.

$m^{-2}\,nm^{-2}\,sr^{-1}$), in contrast to the VPD effect ($\Delta SIF(VPD|SM) = -0.03\,mW\,m^{-2}\,nm^{-2}\,sr^{-1}$). Thus, the comparison of ($\Delta SIF(SM|VPD)$) and $\Delta SIF(VPD|SM)$ enables the disentangling of their relative role in governing dryness stress.

Next, we examine the respective SM and VPD effects on SIF globally. To ensure comparability in space, the SIF time series at each pixel are normalized by the average SIF exceeding the 90th percentile. Temperature and radiation can also limit ecosystem production, therefore, we have filtered out days when other meteorological drivers were likely to be more important than SM or VPD in limiting ecosystem carbon and water fluxes throughout the analyses, following previous studies[12,23]. We find that $\Delta SIF(SM|VPD)$ is negative across most vegetated land areas, robustly indicating the limiting role of low SM to SIF (Fig. 4a, b)

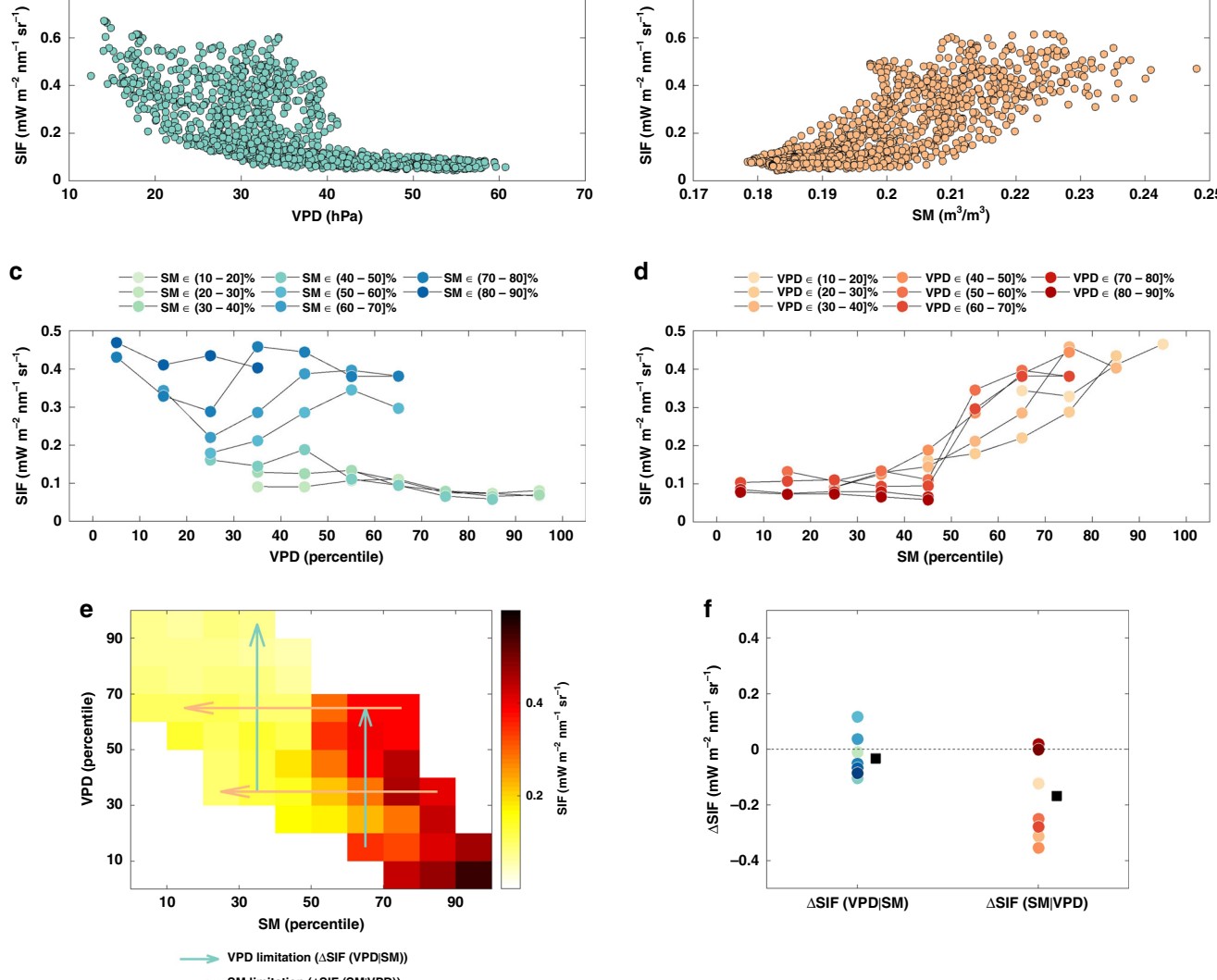

**Fig. 3 Disentangling soil moisture and vapor pressure deficit limitation effects. a** Daily solar-induced chlorophyll fluorescence (SIF) versus daily vapor pressure deficit (VPD). **b** Daily SIF versus daily soil moisture (SM). **c** Daily SIF versus daily VPD, binned by SM. **d** Daily SIF versus daily SM, binned by VPD. **c**, **d** circles denote the averaged SIF within each bin of VPD and SM. **e** Average SIF in each percentile bin of SM and VPD. The cyan arrows indicate the VPD limitations on SIF without SM-VPD coupling (ΔSIF(VPD|SM)), and the orange arrows indicate the SM limitations on SIF without SM-VPD coupling (ΔSIF (SM|VPD)). For better readability, only four arrows are shown. **f** Distribution of ΔSIF(VPD|SM) and ΔSIF(SM|VPD). Circles denote the ΔSIF(VPD|SM) and ΔSIF(SM|VPD) in each bin. Squares denote the corresponding mean. The example pixel is located in Mali, West Africa at 14.25°N, −4.75°E. See Methods for more details.

and consistent with plant physiological understanding and previous studies[4,7]. The units refer to the fractions relative to average SIF exceeding the 90th percentile in each grid cell. Large ΔSIF(SM|VPD) are identified in mid-latitudes, including southern North America, central Eurasia, southern Africa, and Australia. In contrast, ΔSIF(VPD|SM) is small and close to 0 across large areas, but it was larger than ΔSIF(SM|VPD) in tropical Africa surrounding the equator (Fig. 4c, d). Globally, a change from the wettest SM to the driest SM under constant VPD reduces SIF by up to 14.9% on average, whereas a change in VPD from lowest to highest quantiles under constant SM has little effect on SIF (−3.8%) on average. Locally, the areas where the strength of SM effects on SIF (|ΔSIF(SM|VPD)|) exceeds that of VPD effects (|ΔSIF(VPD|SM)|) are widespread, which is also visible along the latitudinal gradient (Fig. 4e, f). In total, |ΔSIF (SM|VPD)| is larger than |ΔSIF(VPD|SM)| across 71.3% of land vegetated areas with valid data, by contrast, VPD is more

important than SM in 26.7% of corresponding areas. Furthermore, our findings suggest that many previous estimates of the role of VPD on ecosystem production are likely exaggerated[16,24] as they did not account for the strong SM-VPD coupling as a confounding factor. In boreal and tropical regions, both SM and VPD have little effect on SIF, which is controlled by radiation and temperature[7,25]. The spatial patterns of ΔSIF(SM|VPD)—ΔSIF (VPD|SM) are robust to the choice of the particular forcing data set (Supplementary Figs. 7–11). However, when using the GOME-2 SIF and SCIAMACHY SIF with the local overpass time at 9:30 am and 10:00 am, the VPD effects are weaker than that in CSIF (reducing SIF by 0.1% and 0.02% on average globally), including most of Africa (excluding the Sahara) as well as large areas of central South America, southern Asia, and Australia (Supplementary Figs. 9–11). This raise a caveat that using SIF retrieved in the morning would underestimate the VPD effects. To further test the robustness of our result, we

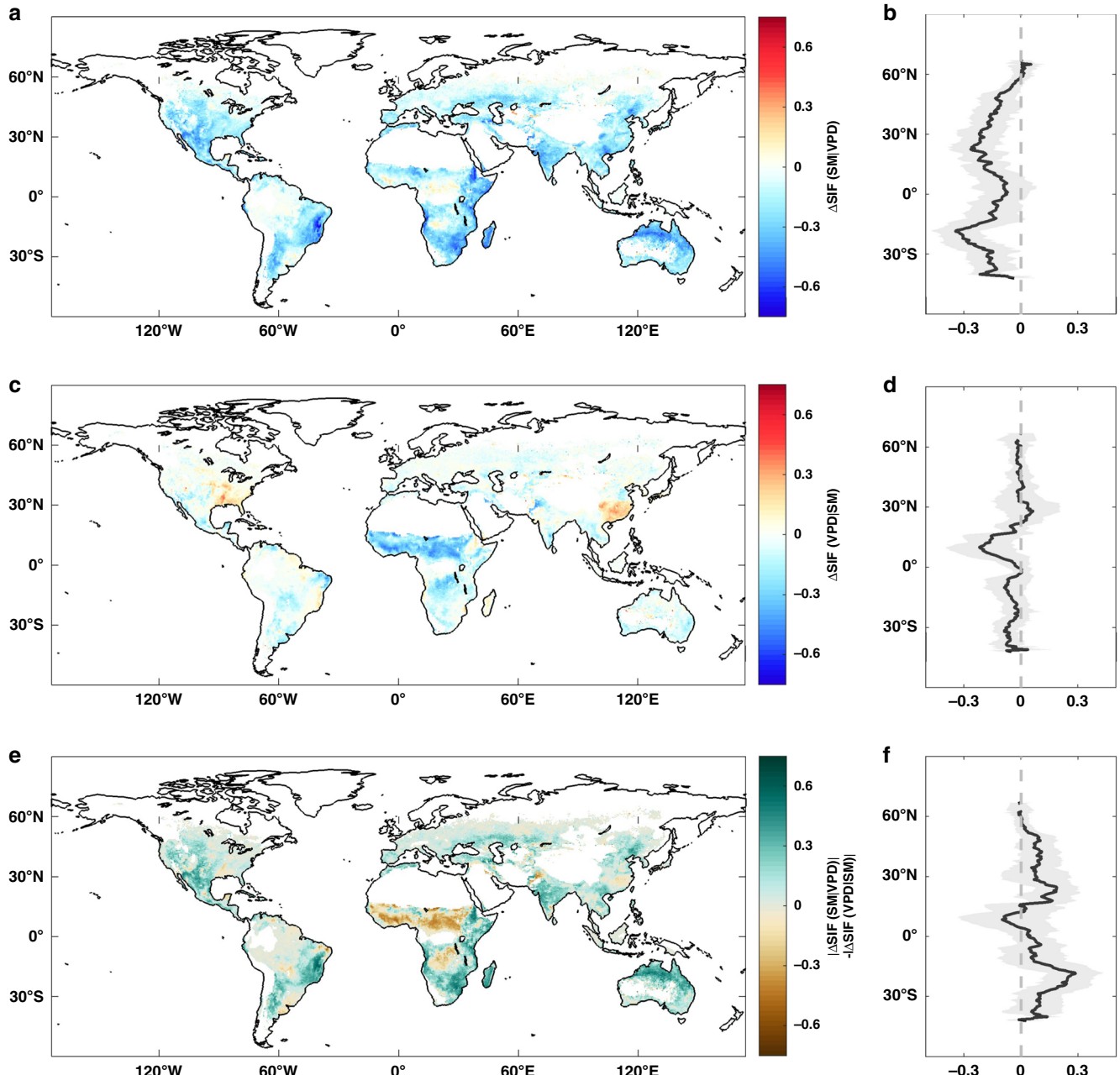

**Fig. 4 Effect of soil moisture and vapor pressure deficit on ecosystem production globally. a**, **c**, **e** Spatial distribution of the changes in solar-induced chlorophyll fluorescence (SIF) caused by low soil moisture (SM) (ΔSIF(SM|VPD)) and high vapor pressure deficit (VPD) (ΔSIF(VPD|SM)), and their differences in absolute values (i.e., |ΔSIF(SM|VPD)|−|ΔSIF(VPD|SM)|). **b**, **d**, **f** Zonal means of SM and VPD effects on SIF and their differences in absolute values. The units refer to the fractions relative to average SIF exceeding the 90th percentile in each grid cell. Black lines indicate the mean values, and gray shaded bands show the standard deviation. Regions with sparse vegetation and regions without valid data are masked in white.

standardized the SIF by photosynthetically active radiation (PAR) to remove possible radiation effects[26], limited the data to a narrow temperature range to remove possible temperature effects and aggregated data to a coarser time resolution or using 20 percentile bins, yielding similar results (Supplementary Figs. 12–15). Thus, we demonstrate that SM is the dominant factor in driving the response of ecosystem production to dryness at the ecosystem scale across most land vegetated areas, except for tropical and boreal areas.

Different from a recent global assessment of SM stress on ecosystem production that estimates the relation between SM stress and background climate from a small sample of flux sites[18], our results build on data with global coverage and hence provide spatially explicit information of SM stress. Further converting the SIF decrease to the actual carbon loss would largely help quantify changes in terrestrial carbon fluxes under drought. Furthermore, our conclusions contradict many laboratory experiments that show strong VPD effects on stomatal conductance at the leaf scale[27,28]. This again indicates that the stomatal sensitivity to VPD do not definitely determine the same VPD response of plant water and carbon fluxes at the ecosystem scale[29,30], but some ecosystem scale measurements reveal that stomatal sensitivity to VPD can matter in some cases[11,12]. Key processes driving the weak plant photosynthesis response to VPD at the ecosystem scale need to be addressed in future work, such as the role of ecosystem water use efficiency, water storage and hydraulic strategies[29].

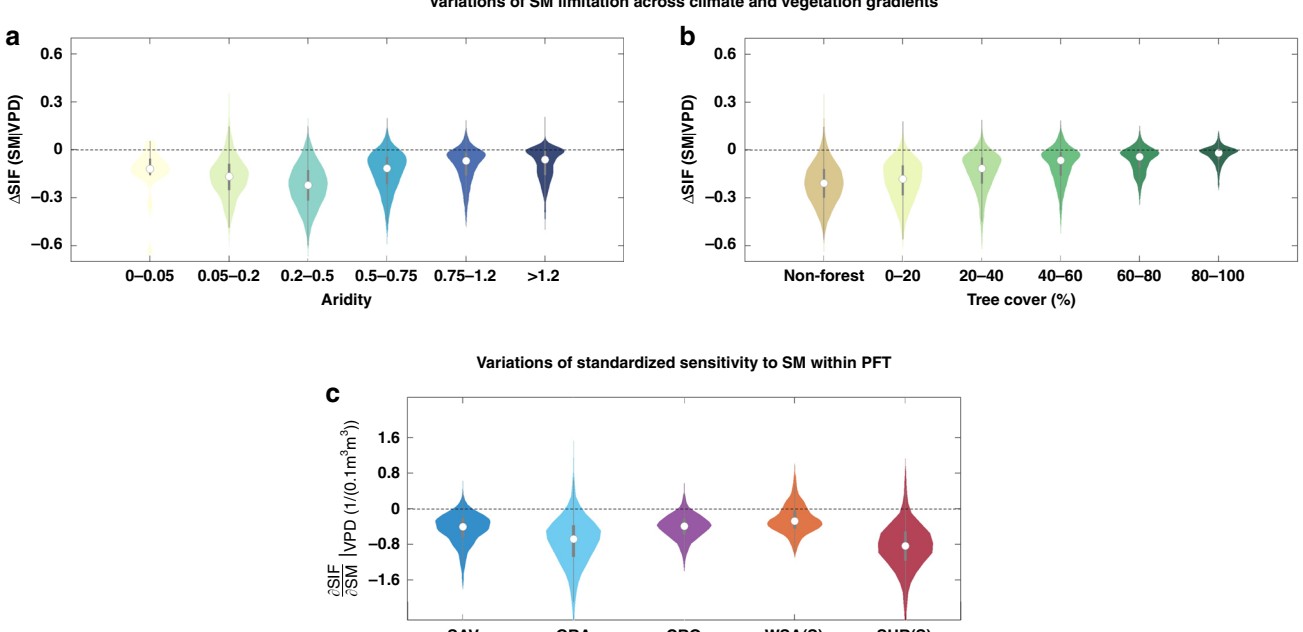

**Fig. 5 Dependence of soil moisture dryness stress on climate and vegetation gradients.** Violin plots of soil moisture (SM) limitation effects ($\Delta$SIF(SM|VPD)) across **a** aridity gradients and **b** tree cover gradients. **c** Violin plots of the sensitivity of solar-induced chlorophyll fluorescence (SIF) to SM (i.e., $\frac{\delta SIF}{\delta SM}|_{VPD}$) within different plant functional types: SHR(S), shrubland (south of 45° N); GRA, grassland; CRO, cropland; WSA(S), woody savanna (south of 45° N); SAV, savanna. White dots indicate the median values, gray boxes cover the interquartile range, and thin gray lines reach the 5th and 95th percentiles.

**Dependence of SM stress on climate and vegetation gradients.** We find that SM limitation effects ($\Delta$SIF(SM|VPD) are largest in semi-arid ecosystems (Fig. 5a), including shrubland, grassland, and savannah ecosystems. These are the ecosystems that are the main drivers of the interannual variability in global terrestrial $CO_2$ flux[31,32]. In contrast, VPD effects are much weaker in these regions (Fig. 4c). This suggests that SM could be more important than VPD in driving interannual variability of global terrestrial carbon uptake. As SM stress is strongest in drylands, the projected expansion of drylands[33] is likely to increase the influence of SM on the future global carbon cycle. In addition, we find that regions with lower tree fraction exhibit a larger response to SM stress globally (Fig. 5b). This is in line with recent findings[34], and further verifies the robustness of our results. Our findings also highlights the differential dryness response of ecosystems along a tree cover gradient.

The representation of dryness stress on plant photosynthetic $CO_2$ assimilation can differ largely between TEMs and is considered one of the largest uncertainties in predicting future land carbon uptake and climate[35–37]. Their representations in TEMs often uses an empirical function that only varies by plant functional type (PFT)[38], which have generally not been validated against observational empirical data. Therefore, we explored the observed standardized sensitivity of SIF to SM. We find that the sensitivity of ecosystem production to changes in SM can vary largely even in the same PFT with strong observed dryness effects (Fig. 5c). This is consistent with recent findings that the grassland's sensitivity to dryness can vary greatly[39]. The differences of dryness response in the same PFT are, e.g., related to plant species, plant height and plant hydraulic processes, such as plasticity variations in xylem and mesophyll conductance, embolism resistance, or water storage[40]. At present, evaluating and incorporating more plant hydraulic processes into the next generation of terrestrial ecosystems is on the way[41]. Our results of dryness effects on ecosystem production thus enables an evaluation of further TEM evolution.

In summary, we provide global results of SM and VPD stress on SIF and demonstrate that SM, rather than VPD, is the dominant driver leading to drought limitation on vegetation productivity at the ecosystem level across most vegetated land areas. VPD stress on ecosystem production is almost lost across large areas without SM-VPD coupling. We thus make the case for revisiting the role of VPD in previous studies that neglected the strong SM-VPD coupling. Furthermore, models that do not correctly disentangle the respective VPD and SM limitations cannot adequately predict the dryness stress on ecosystems and associated rough risks to human well-being. The next challenge is to incorporate the observations to constrain the representation of dryness stress on plants in models, which would also reduce uncertainties in the projection of terrestrial $CO_2$ fluxes and associated climate projections.

## Methods

**SIF**. Chlorophyll fluorescence is the long-wave radiation re-emitted by chlorophyll during photosynthesis. Solar-induced fluorescence (SIF) is therefore mechanistically linked to photosynthesis and is shown to have a near-linear relationship with ecosystem GPP at the ecosystem scale[42,43]. SIF is therefore used as the indicator of GPP in this study. SIF retrieved from three independent missions are used, including OCO-2 (Orbiting Carbon Observatory-2), Global Ozone Monitoring Experiment (GOME-2), and SCIAMACHY (Scanning Imaging Absorption SpectroMeter for Atmospheric Chartography) missions. For OCO-2, the equatorial overpass time is 1:30 pm. Because OCO-2's sampling strategy causes vast spatial gaps between orbits and limits the sampling frequency, the number of observations is not sufficient for our analyses (Supplementary Fig. 17). Therefore, we used a recent spatially continuous OCO-2 SIF data set (CSIF) that fills the spatial gaps by using MODIS surface reflectance and neural networks[22]. The resulting OCO-2 CSIF is estimated at 740 nm and spans from 2000 to 2016, with a spatial resolution of 0.5° × 0.5°. Instantaneous CSIF is demonstrated to well capture the spatial and temporal patterns and variability of original OCO-2 SIF accurately. Independent comparisons with GPP estimates from 40 flux towers demonstrate strong consistency, confirming the effectiveness of CSIF to indicate GPP[22]. However, some uncertainties of CSIF still need to be noted. MODIS surface reflectance data includes some morning observations, possibly bring some biases to instantaneous CSIF. The atmospheric attenuation of SIF signal in cloudy days and canopy structure changes are not well considered and require further improvements[22]. For

GOME-2, the equatorial overpass time is 9:30 am, SIF is estimated at 740 nm and from two approaches: Köhler et al.[44], (referring as GOME-2 GFZ) and Joiner et al.[45], with version 28 (referring as GOME-2 N28). The resulting daily GOME-2 SIF spans from 2007 to 2015, with a spatial resolution of $0.5° × 0.5°$. For SCIA-MACHY, the equatorial overpass time is 10:00 am, SIF is estimated at 740 nm and from the approach of Köhler et al.[44]. The resulting daily SCIAMACHY SIF spans from 2002 to 2012, with a spatial resolution of $1.5° × 1.5°$. Instantaneous SIF can account for the possible impacts from diurnal variations, as morning photosynthesis could be not sensitive to dryness[46]. Daily mean SIF is demonstrated to be more strongly correlated with GPP than instantaneous SIF[42]. The daily mean SIF was converted from the instantaneous SIF at the local overpass time following the method documented in previous studies[47,48]. More details of SIF retrieval can be found in the above references, these data sets have been widely used[3,39,49]. The clear-sky instantaneous CSIF was mainly used due to its validated high quality[22].

**SM**. Because of the lack of global in-situ SM observations, we used daily SM data from reanalysis and satellite retrievals. The reanalysis products are ERA-Interim[50] and Modern-Era Retrospective Analysis for Research and Applications, version 2 (MERRA-2)[51]. The satellite product is the European Space Agency's Climate Change Initiative (ESA CCI), and we used the combined SM data set (v04.4)[52]. For ERA-Interim, with a spatial resolution of ~80 km, the SM content of the soil layers between 0 m and 1 m is summed up (weighted by the thickness of each layer). For MERRA-2, with a spatial resolution of $0.625° × 0.5°$, the root-zone SM content is provided and thus used. For ESA CCI, with a spatial resolution of 0.25°, satellites can only sense the thin (0.5–5 cm) surface soil layer. ERA-Interim SM was used in the main text. All SM data sets were aggregated to a spatial resolution of 0.5°.

**Precipitation, temperature, and radiation**. Daily precipitation and near-surface temperature data were obtained from ERA-Interim or MERRA-2. Daily total surface PAR in all sky conditions at a spatial resolution of 1° was obtained from NASA's Clouds and Earth's Radiant Energy System (CERES), with the version of CERES_SYN1deg_Ed4A. All data sets were aggregated to a spatial resolution of 0.5°.

**VPD**. VPD was calculated as the difference between saturated water vapor pressure, determined by near-surface temperature, and actual water vapor pressure, determined by saturated water vapor pressure and relative humidity. Temperature and relative humidity or specific humidity were obtained from ERA-Interim or MERRA-2. ERA-Interim VPD was used in the main text.

**Aridity index**. The aridity index is defined as the ratio of precipitation to potential evapotranspiration. We used the precipitation and potential evapotranspiration data from the Climate Research Unit v4.01, from 1982 to 2015[53], with a spatial resolution of 0.5°. The classification is provided in Supplementary Table 1[54].

**Tree cover**. Global tree cover was inferred from the global forest change (GFC) v1.6 data set, which was produced from Landsat ETM + time series[55]. Tree cover in GFC was defined as the areal coverage with vegetation canopy height larger than 5 m. The global forest cover in 2009 was used and aggregated from 1 arc-second resolution to 0.5°.

**Vegetation distribution**. MODIS land cover with the classification scheme of the International Geosphere-Biosphere Programme (IGBP) was used. The MODIS IGBP land cover data was obtained from the MCD12Q1 Land Cover Science Data product at a spatial resolution of 0.05°. Moreover, owing to the obvious differences in climate conditions at high and low latitudes, shrubland and woody savanna distributed north and south of 45°N were divided into two categories (Southern and Northern)[31]. Vegetated areas are based on the MODIS land cover data. The PFT was aggregated to a spatial resolution of 0.5° using a majority filter. Access information of all data sets is provided in Supplementary Table 2.

**Analysis**. To investigate the response of vegetation to dryness, we focus on the growing season and days when the SM and VPD effects were most likely to control ecosystem fluxes and screen out days when other meteorological factors were likely to have a larger influence on fluxes. Following previous studies[12,23], for each pixel, we restrict our analyses to the days in which: (i) the daily average temperature >15 °C; (ii) sufficient evaporative demand existed to drive water fluxes, constrained as daily average VPD > 0.5 kPa; (iii) high solar radiation, constrained as daily average photosynthetic photon flux density >500 $\mu mol\ m^{-2}\ s^{-1}$.

Based on the data in the filtered days, for each pixel, we determined the threshold values of 10th, 20th, …, and 90th percentile of SM and VPD, which will then be used to bin the data. Data of all variables (SIF, SM, VPD, and etc.) are sorted into 10 bins according to the 0–10th, 10–20th, …, 80–90th, and 90–100th percentiles of SM or VPD. This binning procedure does not change the temporal match between data (Supplementary Fig. 18). Because SM and VPD are largely decoupled in each SM bin or VPD bin (Fig. 2 and Supplementary Figs. 2–5), we can

disentangle the respective effects of SM and VPD on SIF. For better comparability in space, SIF time series is normalized by the average SIF exceeding 90th percentile per pixel. Next, as the example shown in Fig. 3 in the main text, within each SM bin ($i = 1, 2, …, 10$), the data are further sorted according to VPD, and there are $n_{i,min}$, …, $n_{i,max}$ VPD bins. In particular, $n_{i,min}$ to $n_{i,max}$ is determined by the minimum, maximum VPD value at each SM bin and predetermined VPD threshold values (as illustrated in Fig. 3c, e). In the same way, within each VPD bin ($j = 1, 2, …, 10$), there are $n_{j,min}$, …, $n_{j,max}$ SM bins (as illustrated in Fig. 3d, e). Only bins where >10 data points are available are used in the further analysis. Another example is in Brazil (Supplementary Fig. 19).

The binned averages were used to quantify the limitations of low SM and high VPD to SIF. VPD limitation on SIF without SM-VPD coupling (termed ΔSIF (VPD|SM)) was derived from the changes in SIF from low VPD to high VPD at each SM bin (as illustrated by cyan arrows in Fig. 3e). Here we applied two approaches:

(i)  we calculate the difference between SIF at the highest VPD bin and lowest VPD bin in each SM bin to derive the ΔSIF(VPD|SM), as follows:

$$\Delta SIF(VPD|SM) = \frac{1}{I}\sum_{i=1}^{I} SIF_{i,n_{i,max}} - SIF_{i,n_{i,min}} \quad (1)$$

where $I$ is the number of populated SM bins, $i$ is the specific SM bin number, $n_{i,max}$ and $n_{i,min}$ is the maximum and minimum VPD bin number at SM bin $i$. Equally, SM limitation on SIF without SM-VPD coupling (termed ΔSIF (SM|VPD)) was derived from the changes in SIF from high SM to low SM at each VPD bin (as illustrated by orange arrows in Fig. 3e), as follows:

$$\Delta SIF(SM|VPD) = \frac{1}{J}\sum_{j=1}^{J} SIF_{m_{j,min}\cdot j} - SIF_{m_{j,max}\cdot j} \quad (2)$$

where $J$ is the number of populated VPD bins, $j$ is the specific VPD bin number, $m_{j,min}$ and $m_{j,max}$ is the minimum and maximum SM bin number at the VPD bin $j$. Limited by the small number of valid values in some pixels, $I$ and $J$ can be <10. The response of plant photosynthesis to SM and VPD can be non-linear[3,10], this approach can overcome this limitation. A logarithmic function for VPD is often used to account for the non-linear stomatal response to VPD[10], but the choice of VPD and ln(VPD) would not affect our results. This is because our approach binned data according to quantiles; the data would fall into in the same VPD bins regardless of the choice of VPD and ln(VPD).

(ii) based on binned averages, we fitted a linear regression between SIF and VPD in each SM bin. Consequently, the changes in SIF from lowest VPD bin to highest VPD bin from fitted linear functions were assigned to ΔSIF (VPD|SM). Likewise, SM stress in SIF (ΔSIF(SM|VPD)) was also quantified. This approach can reduce the potential biases caused by extreme values with low data quality in approach (i), but cannot account for non-linear relations. These two approaches lead to similar results (Supplementary Fig. 16), underlining the robustness of our conclusions. We applied approach (i) in the main text. In addition, the change in SIF per change in $0.1\ m^3/m^{-3}$ SM was defined as the sensitivity of SIF to SM, i.e., $\frac{\delta SIF}{\delta SM}|_{VPD}(1/0.1\ m^3/m^{-3})$. This procedure removes the changes in SIF caused by SM range and ensures that the sensitivity of SIF to SM are also comparable in space. Note that we only account for the relatively shallow soil water, whereas deep SM or other types of water storage (e.g., groundwater) may be also relevant for vegetation growth for deep-rooted plants[56], possibly leading to the underestimation of SM effects.

## Data availability
Data supporting the conclusions of this study are properly cited and publicly available. Details are provided in Supplementary Table 2.

## Code availability
The data in this study were analyzed with publicly available tool packages in MATLAB and the figures were produced with MATLAB. All scripts are available upon requests.

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

## Acknowledgements

We acknowledge producers of data sets used in this study. We acknowledge Liangyun Liu for constructive suggestions. L.L., L.G. and S.I.S. acknowledge partial support from the European Union's Horizon 2020 Research and Innovation Program (grant agreement 821003 (4C)). L.L. was also funded by the China Scholar Council fellowship.

## Author contributions

S.I.S. and L.L. designed the study. L.L., S.I.S., L.G. and M.H. performed the research. L.L. carried out the analyses. D.Q. and S.L. contributed to the interpretation of the results. L.L. wrote the paper with contributions from all co-authors.

## Competing interests

The authors declare no competing interests.
