## [Peer Review File · Nature Communications]

Reviewers' comments:

Reviewer #1 (Remarks to the Author):

Liu et al. examine the role of soil moisture vs VPD on ecosystem production. Overall this is a great paper, well-written, and easy to read. My comments are generally directed at the methods with additional minor editorial comments.

1. Generally the title is okay, but the use of "global observations" is only directed towards to SIF, right? Otherwise VPD and SM aren't observed, but rather taken from reanalysis datasets.
2. line 37. "two" instead of "twp"
3. In lines 55-59 the authors give examples of models where dryness stress on photosynthesis is represented as a function of SM or VPD. In the previous sentence, the authors mention how dryness stress on photosynthesis can also be represented as a function of both VPD and SM. Can the authors give examples of TEMS that account for both VPD and SM and how is this done in this example TEM? This could highlight how it is done in TEMS, how it might be wrong, and how the authors are improving this relationship.
4. In the first section of results, the authors discuss yearly correlations. I understand that the correlations exist from yearly to daily, but it's odd the yearly is discussed when it is such a large temporal period with large temporal variation.
5. For Fig 1d it would be interesting to examine areas where the correlation is not high. What is unique about these areas (i.e., physical reason for the low correlation)? The authors do not need to do this for this work, but perhaps a question for future research.
6. Could the physical soil properties (clay content, sand content, etc.) be masking or highlighting correlations? Different soil types will certainly have different soil moisture contents just due to differences in porosity. Is this accounted for?
7. I generally understand the binning procedure, but I do have some questions:
 - 7a. Within each SM bin, the VPD corresponding to the SM bin is further binned based on VPDmin to VPDmax. What dictates the number of VPDmin to VPDmax bins within each SM bin? I don't believe that is mentioned or perhaps I missed this. Vice versa for SMmin to SMax bins, obviously.
 - 7b. When SM and VPD values are binned, is there a temporal mismatch that causes the correlations to be near zero? After binning, if the SM and VPD values are not from the same day or same time period, why should there be a correlation? I would think that the SM and VPD correlations are highly temporal, but please correct me if I am wrong. If I am wrong then I may have misunderstood some of the methodology, so perhaps this should be more clearly written in the supplemental section.
8. Why did the authors select the Mali pixel? Does this pixel show the best example of the methods/results? Please explain why this was chosen and if including another pixel makes sense.
9. I think the authors do a good job of highlighting the deltaSIF results, and these results do highlight the VPD and SM differences. But are these deltaSIF values meaningful for ecosystem production? I believe the authors mention some percent changes (13.2% for SIF reduction due to SM) in the results, but I think would be beneficial to highlight the importance of these numbers and what they mean in the grand scheme of climate change or other meaningful examples (e.g., what does 13.2% actually mean for carbon?).
10. The authors compare their results to Novick et al. (2016) and Sulman et al. (2016), both of which

were done at the flux tower scale. The spatial resolution of this study is 0.5 degrees and done with reanalysis/remote sensing datasets, so are these results comparable?

11. The authors should mention the study time period somewhere in the main text.

12. In the Analysis section in the Methods section, the authors mention data restrictions based on temperature, VPD, etc. After these constraints were established, how many data points were left? It would be useful for the reader to know how many data points were used in the analysis, which certainly affects robustness.

Reviewer #2 (Remarks to the Author):

The response of vegetation to dryness is one of the key uncertainties in future climate change predictions, and also governs a range of vegetation management concerns including agricultural yields and forest growth. It also effects a range of ecosystem services. However, the exact sensitivity of vegetation to limitations in available soil moisture vs. atmospheric water demand (e.g. VPD) remains imperfectly understood. This paper re-examines the sensitivity of photosynthesis to these two drivers across the globe using remotely sensed solar-induced fluorescence. The topic is in an important one and the main finding that soil moisture dominates VPD effects is counter to a variety of recent findings as well as long-standing laboratory-based understanding in the field. As such, they are potentially important, but the bar for ensuring robustness is also high. However, there are several major methodological flaws and concerns in this paper that prevent publication.

Firstly, GOME-2 SiF data are known to be extremely noisy at daily timescale, and their use here is problematic. The authors have a valid point that soil moisture and VPD are correlated at coarser timescales, but this does not mean that the SIF data is of sufficient quality to support the analysis or even the approach used here all together. The analysis should be repeated for robustness using SIF data at monthly or even bimonthly scale.

This noise problem is exacerbated by the formula used for Delta SIF(SM|VPD) and Delta SIF(VPD|SM), which relies on using only two single soil moisture values per bin. Why not calculate the slope in each bin, instead of the range, so that there is less sensitivity to noise in individual values? If data quality is a concern, the 10 decile bins could easily be converted to bins in 20 percetile increments if the signal is as strong as the authors claim it is. Furthermore, if the VPD and soil moisture distributions have different degrees of skewness, this could also affect the results, because the SIF at maximum VPD and SIF at minimum VPD in a bin could, for example, be closer together because the max and min VPD are closer together in most of the bins, rather than because of the VOD sensitivity. Calculating the slope across the bin rather than just the range of some normalized SIF would also help with this.

Additionally, the author's filtering by limiting temperature and solar radiation to relatively warm and sunny ranges may not capture all the confounding effects of these variables on GPP, as there may still be co-variation between soil moisture and radiation or temperature (or similarly VPD and radiation or temperature) occurring in each bin. Just because radiation and temperature are not limiting does not mean they don't affect photosynthesis. What do these cross-correlations look like in each bin?

Aside from the above methodological concerns, it should also be noted that these results contradict countless laboratory experiments under controlled conditions showing the strong influence of atmospheric dryness on stomatal conductance independent of variations in soil moisture (e.g. dating back to the work of Collatz, Ball, Berry and others in the late 1980's) How can the authors explain this

contradictory result? Such a major discrepancy with prior understanding should be addressed in the manuscript.

Minor comment:

Line 56-60 It is true that the optimal model representation of plant responses to dryness remains uncertain. However, this is as much related to uncertainty in the physiological processes as it is to any uncertainty about soil moisture vs. VPD, specifically. These lines are potentially unclear in that regard. For example, JSBACH does not incorporate a stomatal response to VPD not because there is uncertainty about whether this response exists, but because the optimal model representation that does not double-count exact sensitivities in the presence of soil moisture-VPD correlation is uncertain. Similarly, the MODIS GPP product does not incorporate soil moisture because of a lack of data availability for doing so, not because soil moisture is assumed to be unimportant for GPP.

Reviewer #3 (Remarks to the Author):

This manuscript uses satellite-based SIF measurements combined with global gridded soil moisture and vapor pressure deficit (VPD) to estimate the relative importance of soil moisture and VPD in driving dryness stress in photosynthesis as evidenced by variations in SIF measurements. Previous studies have investigated this question at the scale of individual sites, and the extension of those results to global scales using this type of analysis is novel and important. The soil moisture results seem well supported and based on a sound analysis.

However, I think the VPD portion of the analysis suffers from a critical and perhaps fatal flaw: The SIF measurements used in the study are collected only once per day, at 9:30 in the morning local time. VPD has strong diurnal variations that drive VPD limitation of stomatal conductance and photosynthesis. For example, see Matheny et al. (2014), Figure 1, panel a: VPD at 9 am in this example is at 40% of its daily maximum, and has a minimal effect on latent heat flux. In contrast, at 3 pm in this example, VPD is at its daily maximum and latent heat flux is reduced by about 80%, suggesting a strong control on stomatal conductance and most likely GPP. Because VPD is low in the morning, and limitation of stomatal conductance typically becomes significant in the afternoon, I would be very surprised to observe a significant VPD limitation effect in a dataset of 9:30 am photosynthesis measurements, because all of the observations are taking place under conditions when VPD would almost never be high enough to be limiting. Because the key goal of this study is comparing the relative impacts of VPD and soil moisture variations, this represents a critical flaw. The observations being used are from a time of day when VPD is at its least limiting, biasing the analysis toward a result that finds little VPD limitation. This could explain why this study did not observe significant VPD effects on SIF, in contrast to previous studies (e.g., Novick et al., 2016; Sulman et al., 2016) that observed significant VPD effects on stomatal conductance or photosynthesis using hourly flux data (and taking correlation of VPD and soil moisture into account). It would be enlightening to see what the actual range of VPD was in each grid cell at 9:30 am daily, to see how often it exceeded 2 kPa (which is the level at which both the papers mentioned above suggest that VPD limitation becomes important relative to soil moisture). The binning approach used in the study masks the actual range of variability in the two drivers, raising the possibility that much greater variation was observed in soil moisture relative to VPD. This is important information when assessing the relative importance of the two drivers. While I think the underlying strategy of the analysis is going in the right direction, to yield robust results it would need to include observations from multiple times of day, especially the afternoon time periods when VPD is high. This would allow the analysis to represent the full range of VPD conditions occurring in each grid cell across the sub-daily time scales when VPD variations would be expected to influence leaf physiological performance.

Also, it can be problematic to assume a linear dependence between SIF and GPP. When VPD is low and stomatal conductance is high, SIF may be small because of high photochemical quenching (SIF and photochemistry competing for photon energy). As VPD increases and stomatal conductance decreases, photochemical quenching may decrease, causing SIF to increase. When VPD gets very high, leaves may droop due to loss of turgor pressure, causing SIF to decrease again. This suggests that variations in SIF with changes in VPD at only one time of day, especially during the morning, may not be an accurate proxy for GPP limitation.

Specific comments:

Line 55: What representation is this statement referring to? Representation in land surface models?

Line 82-83: Since the binning approach is removing most of the variability in one variable, wouldn't correlation coefficient have to decrease as a consequence? I'm not sure this binning approach demonstrates a real decoupling of variability.

Line 97: If anything, this plot suggests that SIF increases with increasing VPD under dry soil conditions, which is counterintuitive. Including some statistical results to back up the qualitative analysis would be helpful.

Line 172: I'm not sure it's accurate to refer to these results as observations when VPD and soil moisture were mostly model-based.

References:

Matheny, A. M., Bohrer, G., Stoy, P. C., Baker, I. T., Black, A. T., Desai, A. R., et al. (2014). Characterizing the diurnal patterns of errors in the prediction of evapotranspiration by several land-surface models: An NACP analysis. *J. Geophys. Res. Biogeosciences* 119, 1458–1473. doi:10.1002/2014JG002623.

Novick, K. A., Ficklin, D. L., Stoy, P. C., Williams, C. A., Bohrer, G., Oishi, A. C., et al. (2016). The increasing importance of atmospheric demand for ecosystem water and carbon fluxes. *Nat. Clim. Chang.* 6, 1023–1027. doi:10.1038/nclimate3114.

Sulman, B. N., Roman, D. T., Yi, K., Wang, L., Phillips, R. P., and Novick, K. A. (2016). High atmospheric demand for water can limit forest carbon uptake and transpiration as severely as dry soil. *Geophys. Res. Lett.* 43, 9686–9695. doi:10.1002/2016GL069416.

We thank the editor and the three reviewers and for their insightful comments and suggestions to the manuscript. The manuscript has been substantially revised following the reviewer comments.

Please see our detailed response below. The original comments are in black, and our responses are given in blue.

Reviewers' comments:

Reviewer #1 (Remarks to the Author):

Liu et al. examine the role of soil moisture vs VPD on ecosystem production. Overall this is a great paper, well-written, and easy to read. My comments are generally directed at the methods with additional minor editorial comments.

Response: We greatly appreciate your encouraging recommendation of our manuscript. We have revised the manuscript according to your comments.

1. Generally the title is okay, but the use of "global observations" is only directed towards to SIF, right? Otherwise VPD and SM aren't observed, but rather taken from reanalysis datasets.

Response: This is correct; we have therefore revised our title to "Soil moisture dominates dryness stress on ecosystem production globally".

2. line 37. "two" instead of "twp"

Response: Corrected. [See line 35]

3. In lines 55-59 the authors give examples of models where dryness stress on photosynthesis is represented as a function of SM or VPD. In the previous sentence, the authors mention how dryness stress on photosynthesis can also be represented as a function of both VPD and SM. Can the authors give examples of TEMS that account for both

VPD and SM and how is this done in this example TEM? This could highlight how it is done in TEMS, how it might be wrong, and how the authors are improving this relationship.

Response: We now mention an example TEM accounts for both VPD and SM as follows: "In contrast, in the TEM G'Day, VPD can limit plant photosynthesis by causing stomatal closure, and SM can constrain plant photosynthesis directly¹⁸". [See lines 57-58]

4. In the first section of results, the authors discuss yearly correlations. I understand that the correlations exist from yearly to daily, but it's odd the yearly is discussed when it is such a large temporal period with large temporal variation.

Response: In the first section of results, we aim to emphasize that the coupling of SM and VPD confounds the ecosystem dryness stress assessment. In this regard, the yearly scale is typically used to represent the condition of strong coupling between SM-VPD across large land areas (Novick et al., 2016). However, the SM-VPD correlation at monthly or daily scale are weaker and not as well-suited to represent confounding effects across large land areas. To clarify this, we have added the following sentence in the revised manuscript as follows: "We display the yearly scale because it is typically used to represent the condition of strong SM-VPD coupling globally¹⁰". [See lines 68-69]

Reference:

Novick, Kimberly A., et al. "The increasing importance of atmospheric demand for ecosystem water and carbon fluxes." *Nature climate change* 6.11 (2016): 1023-1027.

5. For Fig 1d it would be interesting to examine areas where the correlation is not high. What is unique about these areas (i.e., physical reason for the low correlation)? The authors do not need to do this for this work, but perhaps a question for future research.

Response: Following your suggestion, we have added the following statement to the revised manuscript: "There are still low correlations between SIF and SM or VPD in the northern high latitudes or tropical regions, which suggests possible temperature or radiation effects and requires further investigation."

6. Could the physical soil properties (clay content, sand content, etc.) be masking or highlighting correlations? Different soil types will certainly have different soil moisture contents just due to differences in porosity. Is this accounted for?

Response: Yes, we agree that different soil properties/types could potentially impact the correlations. However, this paper focuses on the relative role of soil moisture and VPD in limiting plant photosynthesis. Therefore, further analysis is beyond the scope of this paper and subject to future investigations.

7. I generally understand the binning procedure, but I do have some questions:

7a. Within each SM bin, the VPD corresponding to the SM bin is further binned based on VPD_{min} to VPD_{max}. What dictates the number of VPD_{min} to VPD_{max} bins within each SM bin? I don't believe that is mentioned or perhaps I missed this. Vice versa for SM_{min} to SM_{max} bins, obviously.

Response: Before the binning procedure, we firstly determined the threshold values of the 10th, 20th, ...and 90th percentile of SM and VPD. Then, in each SM bin, the number of VPD_{min} to VPD_{max} bins is determined by the threshold values. The procedure of the determination of SM and VPD threshold values is already described in the 1st version of the manuscript [See lines 263-264]. We also emphasize the role of threshold values as follows: "... the threshold values of SM and VPD, which will then be used to bin the data". [See lines 264] Moreover, we also revised one sentence to clarify this method as follows: "... there are $n_{i,min}$, ..., $n_{i,max}$ VPD bins. In particular, $n_{i,min}$ to $n_{i,max}$ is determined by the minimum, maximum VPD value at each SM bin and predetermined VPD threshold values." [See lines 272-273]

7b. When SM and VPD values are binned, is there a temporal mismatch that causes the correlations to be near zero? After binning, if the SM and VPD values are not from the same day or same time period, why should there be a correlation? I would think that the SM and VPD correlations are highly temporal, but please correct me if I am wrong. If I am wrong then I may have misunderstood some of the methodology, so perhaps this should be more clearly written in the supplemental section.

Response: The SM and VPD are still in temporal match after binning. The binning procedure sorts all variables (SM, VPD, SIF, etc.) into 10 bins on the basis of SM or VPD percentiles (not time); thus, it can cause the “regrouped” data in the bin to be discrete in time but will not change the temporal match between variables. For instance, as illustrated in Figure R1, we first determine the identification numbers of SM data that lies in one SM bin, e.g. the 10th-20th SM. Then, we arrange all variables on these days with the same identify number to one group.

Figure. R1. Illustration of the binning procedure effect on temporal match. The orange dots indicate all data in one SM bin.

We clarify this point in the revised manuscript as follows: “Data of all variables (SIF, SM, VPD, etc.) are sorted into 10 bins according to the 0th-10th, 10th-20th, ..., 80th-90th and 90th-100th percentiles of SM or VPD. This binning procedure does not change the temporal match between data (Supplementary Fig. S17)”. [See lines 264-267]

8. Why did the authors select the Mali pixel? Does this pixel show the best example of the methods/results? Please explain why this was chosen and if including another pixel makes sense.

Response: The key conclusion of this study is that soil moisture dominates the ecosystem production dryness stress at the global scale. The result of the Mali pixel is a typical example to illustrate the conclusion locally. Owing to the large number of pixels, we just show one example pixel, then we show global patterns of our results afterwards. Here we

also show the similar result derived from another pixel located at Brazil (4.25°S, 40.25°W) (Figure. R2).

Figure. R2 Same as Figure3, but using the pixel located at Brazil (4.25°S, 40.25°W).

9. I think the authors do a good job of highlighting the deltaSIF results, and these results do highlight the VPD and SM differences. But are these deltaSIF values meaningful for ecosystem production? I believe the authors mention some percent changes (13.2% for SIF reduction due to SM) in the results, but I think would be beneficial to highlight the importance of these numbers and what they mean in the grand scheme of climate change or other meaningful examples (e.g., what does 13.2% actually mean for carbon?).

Response: Yes, SIF values make sense for ecosystem production. SIF is generally linearly correlated with ecosystem gross primary production (GPP) (Zhang et al.,2016; Li et al., 2018). However, the specific regression coefficients between SIF and GPP remain unresolved globally (Sun et al.,2017), thus it is difficult to mention the actual carbon loss directly. But we would like to further highlight this point in the revised manuscript as

follows: "Further converting the SIF decrease to the actual carbon loss would largely help quantify changes in terrestrial carbon fluxes under drought". [See lines 145-147]

Reference:

Zhang, Yongguang, et al. "Model-based analysis of the relationship between sun-induced chlorophyll fluorescence and gross primary production for remote sensing applications." *Remote Sensing of Environment* 187 (2016): 145-155.

Sun, Ying, et al. "OCO-2 advances photosynthesis observation from space via solar-induced chlorophyll fluorescence." *Science* 358.6360 (2017): eaam5747.

Li, Xing, et al. "Solar- induced chlorophyll fluorescence is strongly correlated with terrestrial photosynthesis for a wide variety of biomes: First global analysis based on OCO- 2 and flux tower observations." *Global change biology* 24.9 (2018): 3990-4008.

10. The authors compare their results to Novick et al. (2016) and Sulman et al. (2016), both of which were done at the flux tower scale. The spatial resolution of this study is 0.5 degrees and done with reanalysis/remote sensing datasets, so are these results comparable?

Response: We acknowledge the different spatial resolution in these studies. The direct quantitative comparison of data is subject to large uncertainties because of the large difference in spatial scale: flux tower footprint about 1km×1km and this study about 55km×55km (0.5° × 0.5°). However, the ecosystem response to dryness stress should be scale-robust here, and we focus on the qualitative comparisons. Therefore, these results are generally comparable.

11. The authors should mention the study time period somewhere in the main text.

Response: We have added a sentence in the main text to highlight the study period as follows: "The study time period mainly spans from 2007 to 2015". [See lines 69-70]

12. In the Analysis section in the Methods section, the authors mention data restrictions based on temperature, VPD, etc. After these constraints were established, how many data

points were left? It would be useful for the reader to know how many data points were used in the analysis, which certainly affects robustness.

Response: Thanks for pointing this out. We have revised the manuscript to show how much data points are left at grid cell as follows: "After data filtering, the average number of data points per $0.5^\circ \times 0.5^\circ$ grid cell in land vegetated areas is about 981 during 2007-2015 (Supplementary Fig. S16)". [See lines 260-261]

Figure R3. Global maps of the average number of data points after data filtering at $0.5^\circ \times 0.5^\circ$.

Reviewer #2 (Remarks to the Author):

The response of vegetation to dryness is one of the key uncertainties in future climate change predictions, and also governs a range of vegetation management concerns including agricultural yields and forest growth. It also effects a range of ecosystem services. However, the exact sensitivity of vegetation to limitations in available soil moisture vs. atmospheric water demand (e.g. VPD) remains imperfectly understood. This paper re-examines the sensitivity of photosynthesis to these two drivers across the globe using remotely sensed solar-induced fluorescence. The topic is in an important one and the main finding that soil moisture dominates VPD effects is counter to a variety of recent findings as well as long-standing laboratory-based understanding in the field. As such, they are potentially important, but the bar for ensuring robustness is also high. However, there are several major methodological flaws and concerns in this paper that prevent publication.

Response: We greatly appreciate the positive verdict of the relevance of our study and the encouraging evaluation of our manuscript. We have revised the manuscript according to your comments.

Firstly, GOME-2 SiF data are known to be extremely noisy at daily timescale, and their use here is problematic. The authors have a valid point that soil moisture and VPD are correlated at coarser timescales, but this does not mean that the SIF data is of sufficient quality to support the analysis or even the approach used here all together. The analysis should be repeated for robustness using SIF data at monthly or even bimonthly scale.

This noise problem is exacerbated by the formula used for $\Delta \text{SIF}(\text{SM}|\text{VPD})$ and $\Delta \text{SIF}(\text{VPD}|\text{SM})$, which relies on using only two single soil moisture values per bin. Why not calculate the slope in each bin, instead of the range, so that there is less sensitivity to noise in individual values? If data quality is a concern, the 10 decile bins could easily be converted to bins in 20 percentile increments if the signal is as strong as the authors claim it is. Furthermore, if the VPD and soil moisture distributions have different degrees of skewness, this could also affect the results, because the SIF at maximum VPD and SIF at minimum VPD in a bin could, for example, be closer together because the max and min VPD are closer together in most of the bins, rather than because of the VOD sensitivity.

Calculating the slope across the bin rather than just the range of some normalized SIF would also help with this.

Response: We understand that the data quality of daily GOME-2 SIF might impact the robustness of our results. As pointed out by the reviewer, we performed our analyses at a short time scale (daily) to ensure the decoupling of soil moisture (SM) and VPD. So that we can disentangle the respective effects of SM and VPD in limiting SIF. To test the robustness of our results to the SIF data, we tested several independent datasets and approaches:

(1) First, we check other available SIF datasets with good quality. One alternative dataset is Orbiting Carbon Observatory-2 (OCO-2) SIF. OCO-2 SIF is known for the fine spatial resolution (1.3 km × 2.25 km) and improved retrieval accuracy, allowing direct validation against flux tower observations (Sun et al., 2017). Indeed, OCO2-SIF was recently reported to strongly correlate with flux tower GPP at daily scale across a wide variety of biomes (Sun et al., 2017; Wood et al., 2017; Verma, et al., 2017; Li et al., 2018a; Li et al., 2018b). However, the spatially sparse nature of OCO-2 SIF limits its use at the global scale, thus it is recommended to merge OCO2-SIF soundings with other spatially continuous satellite observations to generate a spatially continuous dataset (Li et al., 2018b). Fortunately, the SIF community recently developed a global spatially contiguous OCO-2 SIF dataset using MODIS surface reflectance and neural networks and validated the dataset against flux towers (Zhang et al., 2018). This dataset has a temporal resolution of 4 days and a spatial resolution of 0.5°×0.5°. We used this dataset to repeat our analyses and found our results to be robust (Figure. R1).

Figure R1. Same as Figure 4, but using the daily OCO-2 SIF from Zhang et al., 2018.

In addition, we also use an independent new SIF dataset from the Scanning Imaging Absorption SpectroMeter for Atmospheric Chartography (SCIAMACHY) satellite instruments, which is on board European Space Agency's Environmental Satellite (Köhler et al., 2015). This dataset has a temporal resolution of 1 day and a coarser spatial resolution of $1.5^\circ \times 1.5^\circ$. We thus use the independent SCIAMACHY SIF dataset to perform our analyses again and find our results to be robust (Figure. R2).

Figure R2. Same as Figure 4, but using daily SCIAMACHY SIF.

(2) Second, in addition to using GOME-2 SIF from the GFZ center in the 1st version of the manuscript, we also use the latest GOME-2 N28 SIF from NASA (Joiner et al., 2013). Different from GOME-2 GFZ SIF, the GOME-2 N28 SIF used a narrower fitting window that eliminates larger H₂O absorption and new cloudy scenes filtering and bias correction scheme (Joiner, et al., 2016). In a recent SIF dataset comparison study, Bacour et al., 2019 showed that the daily GOME-2 N28 SIF is highly consistent with validated daily OCO-2 SIF in terms of magnitude and temporal variations. To minimize the possible biases from low data quality observations, we only retain the observations that pass the various quality control checks and meet the best quality standard. As a result, we filtered out about 60% of observations, and the average number of individual observations per year and per 0.5° × 0.5° grid cell in land vegetated areas is about 109 (Figure.R3).

Figure R3. Global maps of the average number of individual observations for the GOME-2 N28 SIF, (a) without data quality control and (b) with best data quality control within a year at $0.5^\circ \times 0.5^\circ$.

Then, we again performed our analyses using the data from GOME-2 v28 SIF that passes the best quality control checks, and found our result to be robust (Figure. R4). Other SIF datasets, like TROPOMI SIF or TanSat SIF are not suitable for the analysis because of the short data record or spatial gaps (Mohammed et al., 2019).

Figure. R4 Same as Figure 4, but using GOME-2 N28 SIF with best data quality control.

(3) Third, to further test the robustness of our result to the data quality, we try to aggregate the data to coarser time resolution, under the condition of ensuring SM-VPD decoupling and enough data points for our analyses. After several tests, we aggregated data to 8-days and perform our analyses again using OCO-2 SIF from Zhang et al, 2018. As shown in Figure R5, the results are robust, however, some pixels are lost due to insufficient data points.

Figure. R5 Same as Figure 4, but using the aggregated 8-days OCO-2 SIF.

(4) Fourth, we agree that using the slope in each bin, instead of the range, can reduce the possible noise and account for the skewness of the distribution. Actually, in the 1st version of the manuscript, we applied this approach to certify the robustness (Lines 227-282 and Supporting Fig. S9 in the 1st manuscript version). Here, we applied the slope approach again but using the OCO-2 SIF dataset from Zhang et al., 2018, and certify the robustness of our results again (Figure R6).

Figure. R6 Same as Figure 4, but using the OCO-2 SIF.

Additionally, following your suggestion, we also try to convert the 10 percentile bins to 20 percentile bins to test the robustness our results (Figure. R7) .

Figure. R7 Same as Figure 4, but using the OCO-2 SIF

and 20 percentile bins.

In summary, to test the robustness of our results, we try our best by using three independent SIF datasets and two different approaches and show that our results are robust. We have added these results to the revised manuscript. [See Supplementary Figures S7-S15]

References:

- Köhler, Philipp, Luis Guanter, and Joanna Joiner. "A linear method for the retrieval of sun-induced chlorophyll fluorescence from GOME-2 and SCIAMACHY data." (2015).
- Li, Xing, Jingfeng Xiao, and Binbin He. "Chlorophyll fluorescence observed by OCO-2 is strongly related to gross primary productivity estimated from flux towers in temperate forests." *Remote Sensing of Environment* 204 (2018a): 659-671.
- Li, Xing, et al. "Solar- induced chlorophyll fluorescence is strongly correlated with terrestrial photosynthesis for a wide variety of biomes: First global analysis based on OCO- 2 and flux tower observations." *Global change biology* 24.9 (2018b): 3990-4008.
- Sun, Ying, et al. "OCO-2 advances photosynthesis observation from space via solar-induced chlorophyll fluorescence." *Science* 358.6360 (2017): eaam5747.
- Zhang, Yao, et al. "A global spatially contiguous solar-induced fluorescence (CSIF) dataset using neural networks." *Biogeosciences* 15.19 (2018b): 5779-5800.
- Wood, Jeffrey D., et al. "Multiscale analyses of solar- induced florescence and gross primary production." *Geophysical Research Letters* 44.1 (2017): 533-541.
- Verma, Manish, et al. "Effect of environmental conditions on the relationship between solar- induced fluorescence and gross primary productivity at an OzFlux grassland site." *Journal of Geophysical Research: Biogeosciences* 122.3 (2017): 716-733.
- Joiner, J., et al. "Global monitoring of terrestrial chlorophyll fluorescence from moderate spectral resolution near-infrared satellite measurements: Methodology, simulations, and application to GOME-2." *Atmospheric Measurement Techniques* 6.2 (2013): 2803-2823.
- Joiner, Joanna, et al. "New methods for the retrieval of chlorophyll red fluorescence from hyperspectral satellite instruments: simulations and application to GOME-2 and SCIAMACHY." *Atmospheric Measurement Techniques* 9.8 (2016).

Bacour, C., et al. "Differences Between OCO- 2 and GOME- 2 SIF Products From a Model- Data Fusion Perspective." *Journal of Geophysical Research: Biogeosciences* 124.10 (2019): 3143-3157.

Mohammed, Gina H., et al. "Remote sensing of solar-induced chlorophyll fluorescence (SIF) in vegetation: 50 years of progress." *Remote sensing of environment* 231 (2019): 111177.

Additionally, the author's filtering by limiting temperature and solar radiation to relatively warm and sunny ranges may not capture all the confounding effects of these variables on GPP, as there may still be co-variation between soil moisture and radiation or temperature (or similarly VPD and radiation or temperature) occurring in each bin. Just because radiation and temperature are not limiting does not mean they don't affect photosynthesis. What do these cross-correlations look like in each bin?

Response: We agree that radiation and temperature may have confounding effects on photosynthesis, despite restricting our analysis to small temperature or photosynthetically active radiation (PAR) ranges as much as possible, following previous studies (Sulman et al., 2016; Novick, et al., 2016; Anderegg et al., 2018).

We first tested the correlation between PAR and soil moisture and VPD in bins. As shown in Figure. R8 and R9, PAR is weakly correlated with VPD and SM in each bin. Thus, PAR is unlikely to have confounding effects on photosynthesis.

Fig. R8 Spatial distribution of Pearson's correlation coefficient between SM and PAR (i.e., $r(\text{SM}, \text{PAR})$) in (a-j) 0th-10th, 10th-20th, ..., 80th-90th, and 90th-100th percentiles of VPD. Regions with sparse vegetation and regions without valid data are masked in white.

Fig. R9. Spatial distribution of Pearson's correlation coefficient between VPD and PAR (i.e., $r(\text{VPD}, \text{PAR})$) in (a-j) 0th-10th, 10th-20th, ..., 80th-90th, and 90th-100th percentiles of SM. Regions with sparse vegetation and regions without valid data are masked in white.

However, temperature is inherently correlated with SM because of soil-climate interactions (Figure. R10) (Seneviratne et al., 2010). Also, temperature is also expected to be highly correlated with VPD because the saturation vapor pressure depends exponentially on temperature (Figure. R11).

Fig. R10. Spatial distribution of Pearson's correlation coefficient between SM and temperature (T_a) (i.e., $r(SM, T_a)$) in (a-j) 0th-10th, 10th-20th, ..., 80th-90th, and 90th-100th percentiles of VPD. Regions with sparse vegetation and regions without valid data are masked in white.

Fig. R11 Spatial distribution of Pearson's correlation coefficient between VPD and T_a (i.e., $r(VPD, T_a)$) in (a-j) 0 th-10th, 10th-20th, ..., 80th-90th, and 90th-100th percentiles of SM. Regions with sparse vegetation and regions without valid data are masked in white.

To test the possible temperature effects, the general solution is to restrict the data analysis to a narrow temperature range, as done in a previous study (Novick et al., 2016). Because, in the narrow and warm temperature range or gradient, temperature is unlikely to bias the respective SM and VPD limitation effects on photosynthesis. Therefore, we performed our analysis in the shallow temperature range of 5°C again and showed that our results are robust despite spatial gaps (Figure. R12). We average the SM and VPD limitation effects from the several narrow temperature bins and find the spatial patterns are quite similar to that do not limit the temperature range (Figure. R13). Moreover, we further use a narrower temperature bin of 3°C and find the spatial patterns are quite similar (Figure. R14). Thus, the temperature and radiation are likely not biasing our conclusions. We have added these results to the revised manuscript. [See Supplementary Figures S11-S12]

Fig. R12. Spatial distribution of the changes in SIF caused by low SM ($\Delta\text{SIF}(\text{SM}|\text{VPD})$) and high VPD ($\Delta\text{SIF}(\text{VPD}|\text{SM})$), and their differences in absolute values (i.e., $|\Delta\text{SIF}(\text{SM}|\text{VPD}) - \Delta\text{SIF}(\text{VPD}|\text{SM})|$) in different narrow temperature ranges of (a, d, g) 15°C-20°C, (b, e, h) 20°C-25°C, (c, f, i) 25°C-30°C. Due to the limited number of valid data points, after binning data according to the specific temperature range, there are insufficient amount of data points in some regions. Regions with sparse vegetation and regions without valid data are masked in white.

Figure. R13. Same as Figure 4, but using the averaged estimates from narrow temperature ranges of 15°C-20°C, 20°C-25°C and 25°C-30°C.

Figure. R14 Same as Figure 4, but using the averaged estimates from narrow temperature ranges of 15°C-18°C, 18°C-21°C, 21°C-24°C, 24°C-27°C and 27°C-30°C.

References:

Anderegg, William RL, et al. "Hydraulic diversity of forests regulates ecosystem resilience during drought." *Nature* 561.7724 (2018): 538-541.

Novick, Kimberly A., et al. "The increasing importance of atmospheric demand for ecosystem water and carbon fluxes." *Nature climate change* 6.11 (2016): 1023-1027.

Sulman, Benjamin N., et al. "High atmospheric demand for water can limit forest carbon uptake and transpiration as severely as dry soil." *Geophysical Research Letters* 43.18 (2016): 9686-9695.

Seneviratne, Sonia I., et al. "Investigating soil moisture–climate interactions in a changing climate: A review." *Earth-Science Reviews* 99.3-4 (2010): 125-161.

Aside from the above methodological concerns, it should also be noted that these results contradict countless laboratory experiments under controlled conditions showing the strong influence of atmospheric dryness on stomatal conductance independent of variations in soil moisture (e.g. dating back to the work of Collatz, Ball, Berry and others in the late 1980's) How can the authors explain this contradictory result? Such a major discrepancy with prior understanding should be addressed in the manuscript.

Response: We agree that the plant stomatal response to VPD at the leaf scale from laboratory experiments is well investigated. However, the response of plant photosynthesis to VPD at the ecosystem scale that account for the strong coupling between VPD and SM remains unclear. Conclusions from the leaf scale are not always linearly scaled up to the ecosystem scale, because the processes at the ecosystem scale or larger scales are very complex. Moreover, the stomatal conductance response to VPD do not definitely determine the same VPD response of plant water and carbon fluxes. For instance, there are fundamental differences in the plant transpiration response to rising VPD at the ecosystem scale. Some earlier work has shown that higher VPD tends to increase transpiration rates and remain high for a wide range of species (Monteith 1995; Pataki et al., 1998; O'Grady et al.,1999; Meinzer 2003). This is contrasted by other studies that show evidence of reduced transpiration with high VPD (Farquhar 1978; Franks et al., 1997; Whitley et al., 2013). The Free air humidity manipulation (FAHM) experiment in south-eastern Estonia uniquely manipulates VPD independent from other environmental drivers and reveals that high VPD results in increase in transpiration and aboveground growth and photosynthesis

in forests (Kupper et al., 2011; Oksanen et al., 2018). Besides, plant adaptation strategies at the ecosystem scale may play important role in regulating the exchange of carbon and water under high atmospheric dryness (Kozlowski et al., 2002; Oksanen et al., 2018). For instance, the degree of ecosystem water use efficiency, water storage and hydraulic strategies to mediate the VPD effect remains unclear globally.

As pointed by the reviewer, the novelty of this study lies in using the recently available direct observations of plant carbon uptake at the ecosystem scale to examine the sensitivity to soil moisture and VPD globally. Further investigation of key processes driving ecosystem production responses to VPD beyond the scope of this study and need to be addressed in future work. In summary, in the revised manuscript, we highlighted this point and added the possible explanation as follows: "Our conclusions contradict many laboratory experiments that show strong VPD effects on stomatal conductance at the leaf scale^{26,27}. This again indicates that the stomatal sensitivity to VPD do not definitely determine the same VPD response of plant water and carbon fluxes at the ecosystem scale^{28,29}. Key processes driving the weak plant photosynthesis response to VPD at the ecosystem scale need to be addressed in future work, such as the role of ecosystem water use efficiency, water storage and hydraulic strategies²⁹." [See lines 148-154]

References:

Ball, J. Timothy, Ian E. Woodrow, and Joseph A. Berry. "A model predicting stomatal conductance and its contribution to the control of photosynthesis under different environmental conditions." *Progress in photosynthesis research*. Springer, Dordrecht, 1987. 221-224.

Collatz, G. James, et al. "Physiological and environmental regulation of stomatal conductance, photosynthesis and transpiration: a model that includes a laminar boundary layer." *Agricultural and Forest meteorology* 54.2-4 (1991): 107-136.

Monteith, J. L. "A reinterpretation of stomatal responses to humidity." *Plant, Cell & Environment* 18.4 (1995): 357-364.

Pataki, D. E., et al. "Canopy conductance of *Pinus taeda*, *Liquidambar styraciflua* and *Quercus phellos* under varying atmospheric and soil water conditions." *Tree Physiology* 18.5 (1998): 307-315.

O'Grady, A. P., D. Eamus, and L. B. Hutley. "Transpiration increases during the dry season: patterns of tree water use in eucalypt open-forests of northern Australia." *Tree physiology* 19.9 (1999): 591-597.

Meinzer, Frederick C. "Functional convergence in plant responses to the environment." *Oecologia* 134.1 (2003): 1-11.

Kupper, Priit, et al. "An experimental facility for free air humidity manipulation (FAHM) can alter water flux through deciduous tree canopy." *Environmental and Experimental Botany* 72.3 (2011): 432-438.

Farquhar, Graham D., and Klaus Raschke. "On the resistance to transpiration of the sites of evaporation within the leaf." *Plant Physiology* 61.6 (1978): 1000-1005.

Franks, P. J., I. R. Cowan, and G. D. Farquhar. "The apparent feedforward response of stomata to air vapour pressure deficit: information revealed by different experimental procedures with two rainforest trees." *Plant, Cell & Environment* 20.1 (1997): 142-145.

Whitley, Rhys, et al. "Developing an empirical model of canopy water flux describing the common response of transpiration to solar radiation and VPD across five contrasting woodlands and forests." *Hydrological processes* 27.8 (2013): 1133-1146.

Kozlowski, T. T., and S. G. Pallardy. "Acclimation and adaptive responses of woody plants to environmental stresses." *The botanical review* 68.2 (2002): 270-334.

Oksanen, Elina, et al. "Northern forest trees under increasing atmospheric humidity." *Progress in Botany* Vol. 80. Springer, Cham, 2018. 317-336.

Minor comment:

Line 56-60 It is true that the optimal model representation of plant responses to dryness remains uncertain. However, this is as much related to uncertainty in the physiological processes as it is to any uncertainty about soil moisture vs. VPD, specifically. These lines are potentially unclear in that regard. For example, JSBACH does not incorporate a stomatal response to VPD not because there is uncertainty about whether this response exists, but because the optimal model representation that does not double-count exact sensitivities in the presence of soil moisture-VPD correlation is uncertain. Similarly, the MODIS GPP product does not incorporate soil moisture because of a lack of data availability for doing so, not because soil moisture is assumed to be unimportant for GPP.

Response: Thanks for pointing this out. We revised these sentences as follows: "As a consequence, in combination with the uncertainty in physiological process understanding, the dryness stress on photosynthesis is either represented as a function of SM only^{12,13}, VPD only¹⁴⁻¹⁶ or both¹⁷ in terrestrial ecosystem models (TEMs) and satellite models. For instance, the TEM JSBACH does not incorporate a stomatal response to VPD¹⁸, because it is uncertain if the SM-VPD correlation will cause a double counting of the dryness sensitivity". [See lines 53-58] We removed the example of MODIS GPP product.

Reviewer #3 (Remarks to the Author):

This manuscript uses satellite-based SIF measurements combined with global gridded soil moisture and vapor pressure deficit (VPD) to estimate the relative importance of soil moisture and VPD in driving dryness stress in photosynthesis as evidenced by variations in SIF measurements. Previous studies have investigated this question at the scale of individual sites, and the extension of those results to global scales using this type of analysis is novel and important. The soil moisture results seem well supported and based on a sound analysis.

Response: We greatly appreciate your encouraging evaluation of our manuscript.

However, I think the VPD portion of the analysis suffers from a critical and perhaps fatal flaw: The SIF measurements used in the study are collected only once per day, at 9:30 in the morning local time. VPD has strong diurnal variations that drive VPD limitation of stomatal conductance and photosynthesis. For example, see Matheny et al. (2014), Figure 1, panel a: VPD at 9 am in this example is at 40% of its daily maximum, and has a minimal effect on latent heat flux. In contrast, at 3 pm in this example, VPD is at its daily maximum and latent heat flux is reduced by about 80%, suggesting a strong control on stomatal conductance and most likely GPP. Because VPD is low in the morning, and limitation of stomatal conductance typically becomes significant in the afternoon, I would be very surprised to observe a significant VPD limitation effect in a dataset of 9:30 am photosynthesis measurements, because all of the observations are taking place under conditions when VPD would almost never be high enough to be limiting. Because the key goal of this study is comparing the relative impacts of VPD and soil moisture variations, this represents a critical flaw. The observations being used are from a time of day when VPD is at its least limiting, biasing the analysis toward a result that finds little VPD limitation. This could explain why this study did not observe significant VPD effects on SIF, in contrast to previous studies (e.g., Novick et al., 2016; Sulman et al., 2016) that observed significant VPD effects on stomatal conductance or photosynthesis using hourly flux data (and taking correlation of VPD and soil moisture into account). It would be enlightening to see what the actual range of VPD was in each grid cell at 9:30 am daily, to see how often it exceeded 2 kPa (which is the level at which both the papers mentioned above suggest

that VPD limitation becomes important relative to soil moisture). The binning approach used in the study masks the actual range of variability in the two drivers, raising the possibility that much greater variation was observed in soil moisture relative to VPD. This is important information when assessing the relative importance of the two drivers. While I think the underlying strategy of the analysis is going in the right direction, to yield robust results it would need to include observations from multiple times of day, especially the afternoon time periods when VPD is high. This would allow the analysis to represent the full range of VPD conditions occurring in each grid cell across the sub-daily time scales when VPD variations would be expected to influence leaf physiological performance.

Response: We agree that there is a temporal mismatch between instantaneous SIF and VPD, and VPD diurnal variations could have differential effects on stomatal conductance or photosynthesis. To test the robustness of our results, we performed two approaches:

(1) Following your suggestions, we include SIF observations from the afternoon to test the robustness of our conclusions. We check the current available SIF data with local overpass time in the afternoon. OCO-2, TROPOMI and TanSat overpass at 13:30 local time. However, the length of TROPOMI and TanSat SIF record is too short for our analysis, because TROPOMI and TanSat were launched in October 2017 and December 2016, respectively. OCO-2 SIF has good data quality and a relatively long record. In particular, because of OCO-2's high spatial resolution ($\sim 1.3 \text{ km} \times 2.25 \text{ km}$), OCO-2 SIF allows direct validation against ground-based and airborne SIF observations. Actually, it was recently reported that OCO-2 SIF correlated very well with flux tower GPP in almost all biomes at daily scale from 64 flux sites (Li et al., 2018). Since our study focuses on the global scale, we use a new dataset created by Zhang et al., (2018b). This dataset filled the OCO-2 observational gaps in space and derived spatially contiguous SIF, with the use of MODIS surface reflectance and neural networks. This dataset has a temporal resolution of 4 days and a spatial resolution of 0.5° . Note that SM and VPD are also largely decoupled in 4-day bins (Figure. R1 and R2), which permit disentangling the relative role of SM and VPD in limiting ecosystem production. Then, we performed our analysis again with SIF observations from the afternoon (OCO-2), and found our results are robust (Figure. R3).

Figure. R1 Same as Figure S2, but using 4-day bins.

Figure. R2 Same as Figure S3, but using 4-day bins.

Figure. R3 Same as Figure 4, but using daily instantaneous OCO-2 SIF.

(2) As pointed by the reviewer, many SIF satellite sensors (like GOME-2, OCO-2) just overpass once in a day, because they are inevitably limited by costs and technology. To overcome this temporal mismatch, a daily conversion factor was developed to convert instantaneous SIF at the local overpass time to daily mean SIF, which was found to perform well (Sun et al., 2017; Zhang et al., 2018a; Li et al., 2018). For example, Sun et al., (2017) and Zhang et al (2018a) both showed that daily mean SIF converted from instantaneous SIF is correlated very well with daily mean/integrated flux tower GPP at different biomes. The daily conversion factor relies mostly on the solar zenith angle, because SIF are driven by the incident solar irradiance which is mostly determined by solar zenith angle. Details can be found in Zhang et al., (2018a). Thus, daily mean SIF and daily mean VPD are temporally matchable. We repeated our analysis using daily mean GOME-2 SIF, and found our results are robust (Figure. R5). Zhang et al., 2018b also applied the daily conversion factor to derive the daily mean SIF. We used the daily mean SIF from Zhang et al., 2018b to perform the analyses and certify the robustness of our results again (Figure. R6).

Figure. R5 Same as Figure 4, but using daily mean GOME-2 SIF.

Figure. R6 Same as Figure 4, but using daily averaged OCO-2 SIF.

Therefore, our conclusions appear to be robust to the overpass time of SIF satellites. This can be also explained by that the instantaneous SIF is generally well correlated with daily mean/integrated GPP at the ecosystem scale, despite the GPP - instantaneous SIF correlation being weaker, and GPP - instantaneous SIF slope is different from that of GPP-daily mean SIF (Zhang et al., 2016, Li et al., 2018). That may be why SIF retrieved from satellite missions at different overpass time can all constrain daily, monthly or yearly integrated GPP well.

In summary, we show that our results are robust to the temporal mismatch of instantaneous SIF observations and daily mean estimates of SM and VPD. To illustrate this issue clearly, we added sentences and these results to the revised manuscript as follows: "We also use the OCO-2 SIF dataset with the local overpass time at 13:30, to test the possible impacts from diurnal variations²⁵, resulting a similar result (Supplementary Fig. S9)." [See lines 134-136]

"Note that, to avoid the temporal mismatch between SIF observation and climate, the daily mean SIF converted from the instantaneous SIF at the local overpass time was used here, following previous studies^{45,46}." [See lines 206-208]

References:

- Zhang, Yongguang, et al. "Model-based analysis of the relationship between sun-induced chlorophyll fluorescence and gross primary production for remote sensing applications." *Remote Sensing of Environment* 187 (2016): 145-155.
- Sun, Ying, et al. "OCO-2 advances photosynthesis observation from space via solar-induced chlorophyll fluorescence." *Science* 358.6360 (2017): eaam5747.
- Li, Xing, et al. "Solar- induced chlorophyll fluorescence is strongly correlated with terrestrial photosynthesis for a wide variety of biomes: First global analysis based on OCO- 2 and flux tower observations." *Global change biology* 24.9 (2018): 3990-4008.
- Zhang, Yao, et al. "On the relationship between sub-daily instantaneous and daily total gross primary production: Implications for interpreting satellite-based SIF retrievals." *Remote sensing of environment* 205 (2018a): 276-289.
- Zhang, Yao, et al. "A global spatially contiguous solar-induced fluorescence (CSIF) dataset using neural networks." *Biogeosciences* 15.19 (2018b): 5779-5800.

Matheny, A. M., Bohrer, G., Stoy, P. C., Baker, I. T., Black, A. T., Desai, A. R., et al. (2014). Characterizing the diurnal patterns of errors in the prediction of evapotranspiration by several land-surface models: An NACP analysis. *J. Geophys. Res. Biogeosciences* 119, 1458–1473. doi:10.1002/2014JG002623.

Also, it can be problematic to assume a linear dependence between SIF and GPP. When VPD is low and stomatal conductance is high, SIF may be small because of high photochemical quenching (SIF and photochemistry competing for photon energy). As VPD increases and stomatal conductance decreases, photochemical quenching may decrease, causing SIF to increase. When VPD gets very high, leaves may droop due to loss of turgor pressure, causing SIF to decrease again. This suggests that variations in SIF with changes in VPD at only one time of day, especially during the morning, may not be an accurate proxy for GPP limitation.

Response: We agree that SIF observations with once in a day are subjected to uncertainties. As showed in the last response, as well as in the response to Reviewer 2, we have certified the robustness of results by: (1) converting instantaneous SIF to daily mean SIF; (2) including SIF observations from afternoon and other independent satellite missions.

Specific comments:

Line 55: What representation is this statement referring to? Representation in land surface models?

Response: We have revised the sentence as follows: "... in terrestrial ecosystem models (TEMs) and satellite models". [See line 55]

Line 82-83: Since the binning approach is removing most of the variability in one variable, wouldn't correlation coefficient have to decrease as a consequence? I'm not sure this binning approach demonstrates a real decoupling of variability.

Response: Soil moisture (SM) and VPD are inherently coupled due to land-climate interactions (Seneviratne, et al., 2010). Our aim was to find a situation that the covariance between SM and VPD is very low, i.e., low SM is not accompanied by high VPD. So that,

we can disentangle the respective effects of SM and VPD in limiting SIF. Therefore, we perform our analysis at the short time scale and use the binning approach.

Reference:

Seneviratne, Sonia I., et al. "Investigating soil moisture–climate interactions in a changing climate: A review." *Earth-Science Reviews* 99.3-4 (2010): 125-161.

Line 97: If anything, this plot suggests that SIF increases with increasing VPD under dry soil conditions, which is counterintuitive. Including some statistical results to back up the qualitative analysis would be helpful.

Response: We have revised a sentence as follows: "..., high VPD does not reduce SIF but even increase SIF a bit under dry conditions". [See lines 100-101]

Line 172: I'm not sure it's accurate to refer to these results as observations when VPD and soil moisture were mostly model-based.

Response: The use of "observations" is mainly directed towards to SIF, despite the reanalysis SM and VPD are generally the "best" dataset we can get now. To avoid the ambiguity, we do not refer to these results as observations in the revised manuscript. We also revised the title to "Soil moisture dominates dryness stress on ecosystem production globally".

References:

Matheny, A. M., Bohrer, G., Stoy, P. C., Baker, I. T., Black, A. T., Desai, A. R., et al. (2014). Characterizing the diurnal patterns of errors in the prediction of evapotranspiration by several land-surface models: An NACP analysis. *J. Geophys. Res. Biogeosciences* 119, 1458–1473. doi:10.1002/2014JG002623.

Novick, K. A., Ficklin, D. L., Stoy, P. C., Williams, C. A., Bohrer, G., Oishi, A. C., et al. (2016). The increasing importance of atmospheric demand for ecosystem water and carbon fluxes. *Nat. Clim. Chang.* 6, 1023–1027. doi:10.1038/nclimate3114.

Sulman, B. N., Roman, D. T., Yi, K., Wang, L., Phillips, R. P., and Novick, K. A. (2016). High atmospheric demand for water can limit forest carbon uptake and transpiration as severely as dry soil. *Geophys. Res. Lett.* 43, 9686–9695. doi:10.1002/2016GL069416.

Reviewer #1 (Remarks to the Author):

I am happy with the authors' responses to my concerns and comments. The responses were clear and thorough. My only comment is that I think the authors should include the Brazil example (my comment #8) in the supplemental materials. The authors do not need to discuss this in the main text, but rather point the readers towards an additional example.

Reviewer #2 (Remarks to the Author):

I am glad to see the editors have performed additional sensitivity analyses. I appreciate the extensive revisions from the authors and it is encouraging the results do not change significantly when CSIF and/or the slope of the SIF/VPD relationship is used. However, although I did not notice it in my original review, I see now that each of the SIF data products is converted to a daily data 'to avoid the temporal mismatch between SIF observation [sic] and climate' (line 238 in the tracked changes manuscript). This would be expected to degrade the VPD-SIF relationship! As reviewer 3 also commented on in the first round of comments, GPP wouldn't be expected to be sensitive to VPD in the early morning (and of course at night when there is little photosynthesis). Because VPD and GPP have very different diurnal cycles, performing this analysis at daily resolution therefore would be expected to blur the true photosynthesis-VPD relationship and significantly weakens the value of the analysis! I don't follow the argument that the daily upscaling is needed to 'avoid a temporal mismatch with the climate data'. ERA-Interim is available at 6-hourly intervals (and ERA-5 Land at even finer hourly intervals), so it should be possible to repeat this analysis at a temporal resolution that more closely matches the actual VPD conditions at the times of day when most photosynthesis is happening. For such an analysis, the GOME-2 overpass time is probably not appropriate. Why not use OCO-2, but without gapfilling (since CSIF uses both Aqua and Terra observations and thus also blurs diurnal cycles)?

As a minor comment, line 230 has a typo in the name "Joiner"

Reviewer #3 (Remarks to the Author):

The revisions do a thorough job of responding to the comments from all reviewers, and I think the addition of alternative SIF datasets for comparison goes a long way to address the major issue of overpass time that I raised in my previous review. I do think the interpretation of these alternative datasets could be strengthened. Specifically, looking at the OCO-2 results with an afternoon overpass time (Figure S13c and S14c), the VPD effect is clearly stronger than in the GOME-2 dataset and appears to be significantly negative across most of Africa (excluding the Sahara) as well as large areas of central South America, southern Asia, and Australia. This is exactly what I would expect from shifting toward a time of day when VPD is likely to be more limiting. Given this result, I don't think it is justified to simply describe it as a "similar result" (line 140) without discussing the greater impact of VPD in this dataset. I would suggest adding a sentence or two describing the differences among the results from the different SIF datasets. In a related area, the response to reviewers states that "the daily conversion factor relies mostly on solar zenith angle". Doesn't this mean that daily SIF estimates would be unable to represent any effects driven by VPD fluctuations during the day, since only zenith angle, and not VPD, is included in the derivation of daily values? This seems like it would undercut the accuracy of using daily values in assessing VPD effects.

Besides suggesting a more accurate description of the OCO-2 results, I don't see any major remaining issues with the manuscript. I do have some specific suggestions for clarifications:

Line 29: Does the 70% number include areas where data was not available (white areas on the

maps)?

Line 128-129 and 132-133: I think these percentages should be reported for all three SIF datasets, particularly since results from the OCO-2 dataset appear to be significantly different and had afternoon overpass times that arguably give a more accurate view of VPD effects.

Line 136: In fact, from the OCO-2 map it appears that VPD was important in areas of tropical Africa

Line 154: This sentence mentions laboratory experiments, but ignores the ecosystem-scale flux measurements, which did take correlation of SM and VPD into account, that these results also contradict (e.g. Novick et al, Sulman et al). Those studies showed that stomatal sensitivity to VPD did in fact determine the responses of plant water and carbon fluxes at the ecosystem scale, at least in some cases. The text should acknowledge that context.

Line 209: Should this be 740 nm instead of mm?

Line 215-216: It's not clear if the daily mean SIF was used for all three datasets, or only for the OCO-2 dataset

Line 217: I suggest providing a justification for why this dataset was chosen as the primary one for the analysis.

Figure 4: It is not clear what the units are for the values in this figure. Are they fractions relative to the 90th percentile of SIF in each grid cell? Or the same SIF values as Figure 3 (mW/m²/nm/sr)?

We thank the editor and the three reviewers for their insightful comments and suggestions to the manuscript. The manuscript has been revised following the reviewer comments.

Please see our detailed response below. The original comments are in black, and our responses are given in blue.

Reviewers' comments:

Reviewer #1 (Remarks to the Author):

I am happy with the authors' responses to my concerns and comments. The responses were clear and thorough. My only comment is that I think the authors should include the Brazil example (my comment #8) in the supplemental materials. The authors do not need to discuss this in the main text, but rather point the readers towards an additional example.

Response: We greatly appreciate your recommendation of our manuscript. We have included the Brazil example in the supplemental material.

Reviewer #2 (Remarks to the Author):

I am glad to see the editors have performed additional sensitivity analyses. I appreciate the extensive revisions from the authors and it is encouraging the results do not change significantly when CSIF and/or the slope of the SIF/VPD relationship is used. However, although I did not notice it in my original review, I see now that each of the SIF data products is converted to a daily data 'to avoid the temporal mismatch between SIF observation [sic] and climate' (line 238 in the tracked changes manuscript). This would be expected to degrade the VPD-SIF relationship! As reviewer 3 also commented on in the first round of comments, GPP wouldn't be expected to be sensitive to VPD in the early morning (and of course at night when there is little photosynthesis). Because VPD and GPP have very different diurnal cycles, performing this analysis at daily resolution therefore would be expected to blur the true photosynthesis-VPD relationship and significantly

weakens the value of the analysis! I don't follow the argument that the daily upscaling is needed to 'avoid a temporal mismatch with the climate data'. ERA-Interim is available at 6-hourly intervals (and ERA-5 Land at even finer hourly intervals), so it should be possible to repeat this analysis at a temporal resolution that more closely matches the actual VPD conditions at the times of day when most photosynthesis is happening. For such an analysis, the GOME-2 overpass time is probably not appropriate. Why not use OCO-2, but without gapfilling (since CSIF uses both Aqua and Terra observations and thus also blurs diurnal cycles)?

Response: We greatly appreciate your recognition of our efforts to revise the manuscript. We agree that using OCO-2 SIF is a direct way, but it is not suited in this study because OCO-2's sampling strategy causes vast spatial gaps between orbits and limits the sampling frequency. For instance, the average sampling frequency for 215 flux tower sites is only 3.21/year (Lu et al., 2018). When daily OCO-2 retrievals that pass the quality criteria (documented in OCO-2 Lite files) are aggregated to $0.5^{\circ} \times 0.5^{\circ}$, the average number of observations is about 6.9/year in land vegetated areas (Figure R1), which is not sufficient for our analyses. We clarify this point in the revised manuscript as follows: "Because OCO-2's sampling strategy causes vast spatial gaps between orbits and limits the sampling frequency, the number of observations is not sufficient for our analyses (Supplementary Fig. S17)." [See lines 201-203]

Figure R1. Global maps of the average number of aggregated OCO-2 SIF observations at $0.5^{\circ} \times 0.5^{\circ}$ per year.

We acknowledge that the MODIS surface reflectance data utilized in CSIF includes some information from the morning, but validation shows that the machine learning-reconstructed instantaneous CSIF can capture the spatial and temporal

patterns and variability of original OCO-2 SIF accurately (Zhang et al., 2018a; Zhang et al.,2020). The machine-learning algorithm is trained on more than 1.8 million paired samples from OCO-2 SIF. Furthermore, independent comparisons of CSIF with 40 flux towers GPP demonstrate strong consistency, confirming the effectiveness of CSIF to indicate GPP (Zhang et al., 2018a). The approach of merging OCO-2 SIF retrievals with MODIS reflectance data to generate continuous SIF is well applied and is proven to perform successfully (Gentine et al.,2018; Yu et al., 2019; Li et al., 2019; Zhang et al., 2020). However, CSIF is still subject to some uncertainties, including the use of morning observations. To clarify this point, we highlight the effectiveness and potential caveats of this dataset in the text as follows: "Instantaneous CSIF is demonstrated to well capture the spatial and temporal patterns and variability of original OCO-2 SIF accurately²¹. Independent comparisons with GPP estimates from 40 flux towers demonstrate strong consistency, confirming the effectiveness of CSIF to indicate GPP²¹. However, some uncertainties of CSIF still need to be noted. MODIS surface reflectance data includes some morning observations, possibly bring some biases to instantaneous CSIF. The atmospheric attenuation of SIF signal in cloudy days and canopy structure changes are not well considered and require further improvements²¹." [See lines 206-212]

Then, following your suggestion, we use the instantaneous CSIF (local time of 13:30) to perform the analysis. SIF at the local time of 13:30 is generally close to the maximum photosynthesis in a day. As shown in Figure R2, SM limitation effects are still much larger than VPD effects across large areas, i.e., the key conclusion that soil moisture dominates the ecosystem production dryness globally remains robust (Figure R2e, f). Similar results are also obtained when repeating this analysis at a 6-hourly scale that closest to 13:30 (Figure R3e, f). But VPD effects estimated from instantaneous CSIF is stronger than that in the GOME-2 SIF (Figure R2c, d and Figure R3c, d), especially across Africa (excluding the Sahara) as well as large areas of South America, and Australia. As noted in the Reviewer 3's 2nd review, this result is in agreement with the expectation, and the impacts of overpass time on VPD effects should be described in the text. Accordingly, we mainly used the CSIF in the revised manuscript and added a section as follows:

“However, when using the GOME-2 SIF and SCIAMACHY SIF with the local overpass time at 9:30 am and 10:00 am, the VPD effects are weaker than that in CSIF (reducing SIF by 0.1% and 0.02% on average globally), including most of Africa (excluding the Sahara) as well as large areas of central South America, southern Asia, and Australia (Supplementary Fig. S9-S11). This raise a caveat that using SIF retrieved in the morning would underestimate the VPD effects.” [See lines 136-141] We deleted the argument that “daily upscaling is needed to avoid a temporal mismatch with the climate data”.

Figure R2. Same as Figure 4, but using instantaneous CSIF.

Figure R3. Same as Figure 4, but using instantaneous CSIF and 6-hourly VPD that close to the local time of 13:30.

Besides, this study premised that SIF is the proxy of GPP. It is known that OCO-2's fine spatial resolution ($\sim 1.3 \text{ km} \times 2.25 \text{ km}$) permits direct validation against flux tower observations (Sun et al., 2018). The first global analysis of OCO-2 SIF versus 64 flux tower sites encompassing eight major biome types suggests that SIF-GPP relationship is better at daily timescale ($R^2=0.72$ for daily mean SIF_{757nm} and daily mean GPP) than at the instantaneous scale ($R^2=0.62$ for instantaneous SIF_{757nm} and instantaneous GPP) (Li et al., 2018). The method that converts instantaneous SIF to daily mean SIF is documented in the OCO-2 SIF Lite product (Frankenberg et al., 2015). Sun et al., (2017) and Zhang et al (2018b) also showed that daily mean SIF is correlated very well with daily mean flux tower GPP at different biomes. Therefore, the daily mean SIF is more reliable to indicate GPP, particularly daily mean GPP. Actually, the instantaneous SIF is also strongly correlated with daily mean GPP at the ecosystem scale, despite the correlation being weaker, and the GPP - instantaneous SIF slope is different from that of GPP-daily mean SIF (Zhang et al., 2016, Zhang et al., 2018b). This verifies that SIF retrieved from satellite missions at different overpass time can all indicate GPP well and are strongly correlated. This underlies the reasonable use of both instantaneous SIF and daily mean SIF. As shown in the last response, our key conclusions are robust to the daily mean SIF from CSIF, GOME-2 v28, GOME-2 N28, and SCIAMACHY SIF.

We have added these results to the revised manuscript and supporting material.

References:

Frankenberg, C., 2015. In: C.I.o. Technology (Ed.), Solar Induced Chlorophyll Fluorescence OCO-2 Lite Files (B7000) User Guide.

Zhang, Yongguang, et al. "Model-based analysis of the relationship between sun-induced chlorophyll fluorescence and gross primary production for remote sensing applications." *Remote Sensing of Environment* 187 (2016): 145-155.

Sun, Ying, et al. "OCO-2 advances photosynthesis observation from space via solar-induced chlorophyll fluorescence." *Science* 358.6360 (2017): eaam5747.

Gentine, P., & Alemohammad, S. H. (2018). Reconstructed solar- induced fluorescence: A machine learning vegetation product based on MODIS surface reflectance to reproduce GOME- 2 solar- induced fluorescence. *Geophysical Research Letters*, 45(7), 3136-3146.

Sun, Ying, et al. "Overview of Solar-Induced chlorophyll Fluorescence (SIF) from the orbiting carbon observatory-2: Retrieval, cross-mission comparison, and global monitoring for GPP." *Remote Sensing of Environment* (2018): 808-823.

Li, Xing, et al. "Solar- induced chlorophyll fluorescence is strongly correlated with terrestrial photosynthesis for a wide variety of biomes: First global analysis based on OCO- 2 and flux tower observations." *Global change biology* 24.9 (2018): 3990-4008.

Zhang, Yao, et al. "A global spatially contiguous solar-induced fluorescence (CSIF) dataset using neural networks." *Biogeosciences* 15.19 (2018a): 5779-5800.

Zhang, Yao, et al. "On the relationship between sub-daily instantaneous and daily total gross primary production: Implications for interpreting satellite-based SIF retrievals." *Remote sensing of environment* 205 (2018b): 276-289.

Lu, Xinchun, et al. "Opportunities and challenges of applications of satellite-derived sun-induced fluorescence at relatively high spatial resolution." *Science of the Total Environment* 619 (2018): 649-653.

Li, Xing, and Jingfeng Xiao. "A Global, 0.05-Degree Product of Solar-Induced Chlorophyll Fluorescence Derived from OCO-2, MODIS, and Reanalysis Data." *Remote Sensing* 11.5 (2019).

Yu, L., et al. "High- Resolution Global Contiguous SIF of OCO- 2." *Geophysical Research Letters* 46.3 (2019): 1449-1458.

Zhang, Y., Parazoo, N. C., Williams, A. P., Zhou, S., & Gentine, P. (2020). Large and projected strengthening moisture limitation on end-of-season photosynthesis. *Proceedings of the National Academy of Sciences*, 117(17), 9216-9222.

As a minor comment, line 230 has a typo in the name "Joiner"

Response: Thanks for pointing this out. Corrected. [See line 214]

Reviewer #3 (Remarks to the Author):

The revisions do a thorough job of responding to the comments from all reviewers, and I think the addition of alternative SIF datasets for comparison goes a long way to address the major issue of overpass time that I raised in my previous review. I do think the interpretation of these alternative datasets could be strengthened. Specifically, looking at the OCO-2 results with an afternoon overpass time (Figure S13c and S14c), the VPD effect is clearly stronger than in the GOME-2 dataset and appears to be significantly negative across most of Africa (excluding the Sahara) as well as large areas of central South America, southern Asia, and Australia. This is exactly what I would expect from shifting toward a time of day when VPD is likely to be more limiting. Given this result, I don't think it is justified to simply describe it as a "similar result" (line 140) without discussing the greater impact of VPD in this dataset. I would suggest adding a sentence or two describing the differences among the results from the different SIF datasets. In a related area, the response to reviewers states that "the daily conversion factor relies mostly on solar zenith angle". Doesn't this mean that daily SIF estimates would be unable to represent any effects driven by VPD fluctuations during the day, since only zenith angle, and not VPD, is included in the derivation of daily values? This seems like it would undercut the accuracy of using daily values in assessing VPD effects. Besides suggesting a more accurate description of the OCO-2 results, I don't see any major remaining issues with the manuscript. I do have some specific suggestions for clarifications:

Response: We greatly appreciate your recognition of our efforts to revise the manuscript. As suggested, we have clarified the differences among the VPD effects from the different SIF datasets as follows: "However, when using the GOME-2 SIF and SCIAMACHY SIF with the local overpass time at 9:30 am and 10:00 am, the VPD effects are weaker than that in CSIF (reducing SIF by 0.1% and 0.02% on average globally), including most of Africa (excluding the Sahara) as well as large areas of central South America, southern Asia, and Australia (Supplementary Fig. S9-S11). This raise a caveat that using SIF retrieved in the morning would underestimate the VPD effects." [See lines 136-141] The mechanism of daily

conversion factor is not the topic of this study and is not discussed here. Please see our responses to your other suggestions below.

Line 29: Does the 70% number include areas where data was not available (white areas on the maps)?

Response: Thanks for pointing this out. We clarified this as follows: "land vegetated areas with valid data".

Line 128-129 and 132-133: I think these percentages should be reported for all three SIF datasets, particularly since results from the OCO-2 dataset appear to be significantly different and had afternoon overpass times that arguably give a more accurate view of VPD effects.

Response: Agreed. We reported the percentages in the revised manuscript.

Line 136: In fact, from the OCO-2 map it appears that VPD was important in areas of tropical Africa.

Response: Yes. We added a sentence to emphasize this as follows: "but it was larger than $\Delta\text{SIF}(\text{SM}|\text{VPD})$ in tropical Africa surrounding the equator". [See lines 125-126]

Line 154: This sentence mentions laboratory experiments, but ignores the ecosystem-scale flux measurements, which did take correlation of SM and VPD into account, that these results also contradict (e.g. Novick et al, Sulman et al). Those studies showed that stomatal sensitivity to VPD did in fact determine the responses of plant water and carbon fluxes at the ecosystem scale, at least in some cases. The text should acknowledge that context.

Response: We revised the text as follows: "but some ecosystem-scale measurements reveal that stomatal sensitivity to VPD can matter in some cases^{10,11}." [See lines 155-156]

Line 209: Should this be 740 nm instead of mm?

Response: Corrected. [See line 213]

Line 215-216: It's not clear if the daily mean SIF was used for all three datasets, or only for the OCO-2 dataset

Response: We clarify this as follows: "For all SIF datasets". [See line 221]

Line 217: I suggest providing a justification for why this dataset was chosen as the primary one for the analysis.

Response: We clarify this as follows: "due to its validated high quality²¹". [See line 224]

Figure 4: It is not clear what the units are for the values in this figure. Are they fractions relative to the 90th percentile of SIF in each grid cell? Or the same SIF values as Figure 3 (mW/m²/nm/sr)?

Response: We clarify this as follows: "The units refer to the fractions relative to average SIF exceeding the 90th percentile in each grid cell". [See line 123]

REVIEWERS' COMMENTS:

Reviewer #2 (Remarks to the Author):

I appreciate the thorough response to my previous comments. The revised manuscript is more clear, and I have no further comments.

Reviewer #3 (Remarks to the Author):

After reading the revised manuscript, I think the authors have done a good job of responding to reviewer comments and I am satisfied that there are no more substantial issues with the manuscript.

I have some suggestions for corrections and clarifications below:

Line 37: mortality, not mortally

Line 95-96: For clarity, specify here that the main SIF dataset was CSIF

Line 105: It would be more accurate to say that the increase in SIF with increasing VPD was under moderate soil moisture conditions. The increase happened at 50-70 percentile soil moisture, not under the driest conditions

Line 106: Specify that this statement is referring to the results for this particular site at this point in the manuscript

Line 112-113: This wording with the parentheses is difficult to follow. I would write it out: "(i) SIF in the maximum VPD bin minus SIF in the minimum VPD bin or SIF in the minimum SM bin minus SIF in the maximum SM bin; (ii) ..."

Line 131: The wording here implies that the map shows actual mean reduction in SIF attributed to SM fluctuations over some averaged time period, but the values that are plotted are actually more complicated than that. Based on the methods, the values are an estimate of the maximum observed reduction due to soil moisture, averaged over VPD bins. The translation to a reduction in SIF over the averaged time period is not very clear. I would reword this sentence more precisely to avoid misinterpretation. Something like: "In the average grid cell, a change from the wettest to the driest soil moisture under constant VPD could reduce SIF by up to 14.9%. By contrast, in the average grid cell a change in VPD from lowest to highest quantiles while holding SM constant had little effect on SIF (-3.8%)."

Line 142: In this sentence it is not clear that CSIF was the primary data set used in the analysis

Line 210: The descriptions of the other SIF data sets specify that SIF was estimated at 740 nm and also specify the approach used (Kohler et al or Joiner et al). Can this information be provided for OCO-2?

Line 226-227: There is no comparison between instantaneous and daily SIF in the supplement, so I'm not sure what this sentence is referring to.

Line 278: Remove "be" from "likely to be have a larger influence"

Figure 2: The white color used to mask invalid data is the same color as zero correlation in the color bar. I would suggest using a different color like gray so that zero correlation can be distinguished from missing data in the maps

Figure 4: The units in these figures are described in a sentence that was added to the text referring to these figures. I would also describe the units in the figure caption to make it easy to understand without looking through the text for the explanation

We thank the editor and the reviewers for their insightful comments and suggestions to the manuscript. The manuscript has been revised following the reviewer comments.

Please see our detailed response below. The original comments are in black, and our responses are given in blue.

Reviewers' comments:

Reviewer #2 (Remarks to the Author):

I appreciate the thorough response to my previous comments. The revised manuscript is more clear, and I have no further comments.

Response: We greatly appreciate your recommendation of our manuscript.

Reviewer #2 (Remarks to the Author):

After reading the revised manuscript, I think the authors have done a good job of responding to reviewer comments and I am satisfied that there are no more substantial issues with the manuscript.

Response: We greatly appreciate your recommendation of our manuscript.

I have some suggestions for corrections and clarifications below:

Line 37: mortality, not mortally

Response: Corrected. [See line 39]

Line 95-96: For clarity, specify here that the main SIF dataset was CSIF

Response: We specify the main SIF dataset at first use. [See lines 78-80]

Line 105: It would be more accurate to say that the increase in SIF with increasing VPD was under moderate soil moisture conditions. The increase happened at 50-70 percentile soil moisture, not under the driest conditions

Response: Corrected. [See line 114]

Line 106: Specify that this statement is referring to the results for this particular site at this point in the manuscript

Response: Corrected. [See lines 115-116]

Line 112-113: This wording with the parentheses is difficult to follow. I would write it out: "(i) SIF in the maximum VPD bin minus SIF in the minimum VPD bin or SIF in the minimum SM bin minus SIF in the maximum SM bin; (ii) ..."

Response: Corrected. [See lines 122]

Line 131: The wording here implies that the map shows actual mean reduction in SIF attributed to SM fluctuations over some averaged time period, but the values that are plotted are actually more complicated than that. Based on the methods, the values are an estimate of the maximum observed reduction due to soil moisture, averaged over VPD bins. The translation to a reduction in SIF over the averaged time period is not very clear. I would reword this sentence more precisely to avoid misinterpretation. Something like: "In the average grid cell, a change from the wettest to the driest soil moisture under constant VPD could reduce SIF by up to 14.9%. By contrast, in the average grid cell a change in VPD from lowest to highest quantiles while holding SM constant had little effect on SIF (-3.8%)."

Response: Corrected. [See lines 139-142]

Line 142: In this sentence it is not clear that CSIF was the primary data set used in the analysis

Response: We clarify it in the text. [See lines 78-80]

Line 210: The descriptions of the other SIF data sets specify that SIF was estimated at 740 nm and also specify the approach used (Kohler et al or Joiner et al). Can this information be provided for OCO-2?

Response: Corrected. [See lines 78-80]

Line 226-227: There is no comparison between instantaneous and daily SIF in the supplement, so I'm not sure what this sentence is referring to.

Response: We have deleted this sentence.

Line 278: Remove "be" from "likely to be have a larger influence"

Response: Corrected. [See line 282]

Figure 2: The white color used to mask invalid data is the same color as zero correlation in the color bar. I would suggest using a different color like gray so that zero correlation can be distinguished from missing data in the maps

Response: Corrected.

Figure 4: The units in these figures are described in a sentence that was added to the text referring to these figures. I would also describe the units in the figure caption to make it easy to understand without looking through the text for the explanation

Response: Corrected. [See lines 525-526]